# *ModernVBERT*: Towards Smaller Visual Document Retrievers

**Paul Teiletche** [1 2] **Quentin Macé** [1 3] **Max Conti** [1] **Antonio Loison** [1]
**Gautier Viaud** [1] **Pierre Colombo** [3 4] **Manuel Faysse** [1 3]

## Abstract

Large-scale document retrieval (search) is key in many modern industrial AI pipelines to ground models with relevant contextual information. Increasingly, Visual Document Retrieval (VDR) models, which directly embed images of document pages, are used as an alternative to text-only retrievers. While these models are historically repurposed generative VLMs fine-tuned for embedding tasks, we revisit this design choice in this paper and systematically develop strong VDR models from the ground up. Through controlled experiments, we isolate the impact of key training factors such as attention masking, multimodal data regimes, and contrastive objectives at all phases of training. Our findings confirm that current VDR performance is constrained by generative modeling, especially in multi-vector settings. Building on these insights, we train *ModernVBERT*, a 250M-parameter vision-language encoder that outperforms recent models up to 10 times its size when fine-tuned on document retrieval tasks. Thanks to its compact design, *ModernVBERT* enables efficient retrieval inference on CPU hardware, while maintaining competitive performance. Models, code and data are available at https://huggingface. co/ModernVBERT.

## 1. Introduction

Retrieval-Augmented Generation (RAG) (Lewis et al., 2020) enhances LLM performance by providing relevant context to the model, reducing hallucinations and inference costs over long context inputs (Gao et al., 2024; Yu

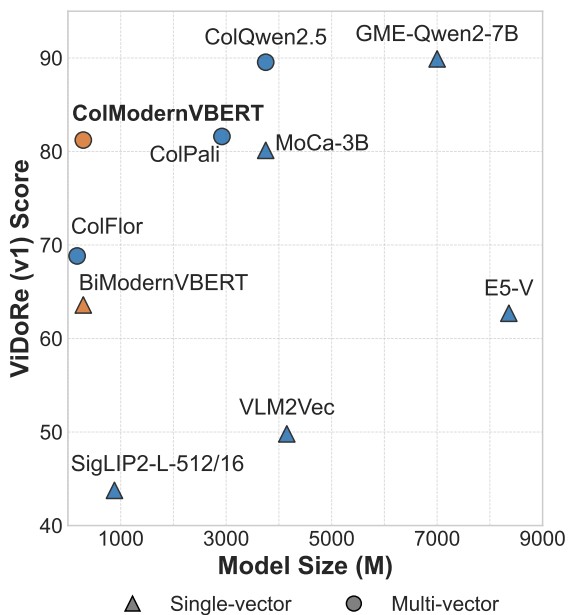

*Figure 1.* **Pareto efficiency.** *ColModernVBERT* outperforms models in its category on ViDoRe, achieving a leading performance-size tradeoff.

et al., 2024; Nguyen et al., 2024; Yu et al., 2025). At its core, RAG relies on neural information retrieval (IR) models to identify contextual information from large-scale collections of documents. The first retrieval stage is crucial for the overall performance of the search pipeline: document recall directly upper-bounds the quality of the generated answers, and its latency partially drives the overall efficiency of the system (Lin & Byrne, 2022). As a result, improving document retrieval—particularly for complex, information-dense formats such as PDFs—is a key lever for increasing the accuracy and cost-effectiveness of industrial RAG deployments.

Historically, document retrieval has relied on extracting text through slow and error-prone preprocessing pipelines (OCR, captionning), and indexing the extracted lossy textual content. Visual Document Retrieval (VDR) recently appeared as a shift toward native multimodal understanding by repurposing large generative vision–language decoders (Alayrac et al., 2022) as retrieval embedding models via post-hoc contrastive fine-tuning (Ma et al., 2024; Faysse

[1]Illuin Technology, Paris, France [2]École Polytechnique Fédérale de Lausanne (EPFL), School of Computer Science, Lausanne, Switzerland [3]MICS, CentraleSupélec, Université Paris-Saclay, France [4]Equall.ai, Paris, France. Correspondence to: Paul Teiletche <paul.teiletche@epfl.ch>.

*Proceedings of the 43rd International Conference on Machine Learning*, Seoul, South Korea. PMLR 306, 2026. Copyright 2026 by the author(s).

et al., 2025; Jiang et al., 2025). Given a user query, VDR models retrieve relevant document pages by matching the query against image-based page representations.

Although VDR simplifies end-to-end pipelines, reduces indexing latency, and improves performance on benchmarks(Guo et al., 2025; Wu et al., 2025), it remains unclear whether its current implementation is optimal. As previous work has shown the limitations of generative modeling in text-retrieval (Lee et al., 2025; Gisserot-Boukhlef et al., 2025), and since VDR relies on generative underlying models, alternative design choices could yield superior performance at comparable budget. Furthermore, scaling trends observed in generative tasks (Wei et al., 2022) are less pronounced for embedding models; while correlated with model size, strong retrieval performance remains attainable with small models (Clavié, 2024; Takehi et al., 2025).

In this work, we systematically study the impact of core design decisions in training visual document retrievers (VDRs). Through controlled experiments–ranging from pretraining objective to domain-specific fine-tuning parameters–we aim to answer a central question: *Which design choices most effectively improve performance in modern visual document retrievers?*

Our results notably indicate that causal attention is suboptimal for document retrieval and that principled design of modality alignment and contrastive training substantially improves performance. We validate these insights by training *ModernVBERT*, a visual document Bert-like encoder model that combines a pretrained text encoder with a vision encoder. The model surpasses the performance of significantly larger VDR models, with drastically reduced online latency. Overall, these findings highlight the benefits of designing specialized visual document retrievers from the ground up. We release the model, checkpoints, and the training code at `https://huggingface.co/ModernVBERT`. Our main contributions are:

**1. Revisiting Core Assumptions in VDR.** We identify key training factors that govern downstream retrieval performance, including the role of token-level training objectives, cross-modal data and the strong influence of attention masking schemes on multi-vector representations.

**2. *ModernVBERT*.** We train and release a small 250M multimodal encoder designed specifically for Visual Document Retrieval. On standard VDR benchmarks, *ColModernVBERT*, the multi-vector variant surpasses models 10x larger its size with much improved latencies.

*Table 1.* **Ablation models.** All models are trained on the same data with identical architecture and optimization; only the attention masking and associated language modeling objective differ. (*) For `dec-to-enc`, the transition applies only during text pretraining; when training the ablation model, we use MLM.

| Model | Attention mask / LM objective |
|---|---|
| `enc` | Bidirectional / MLM |
| `dec` | Causal / CLM |
| `dec-to-enc` | Causal $\rightarrow$ Bidirectional / CLM $\rightarrow$ MLM* |

## 2. Methodology

Our analysis aims at quantifying the impact of VDR training design decisions. Contrary to previous work, we begin our analysis as early as language model modality alignment and iteratively study design choices by modifying design choices independently to reduce confounding factors as much as possible (Allen-Zhu & Li, 2025).

**Controlled Experimental Setup.** A central point of interest is the impact of causal versus bidirectional attention masks. While this question has recently been studied in the context of textual representations (Gisserot-Boukhlef et al., 2025; Weller et al., 2025), we extend the analysis to the vision modality. We use checkpoints released by Gisserot-Boukhlef et al. (2025), consisting of a series of closely matched 210M-parameter transformer models based on the Llama architecture (Touvron et al., 2023). All models are trained on the same 100B tokens and are identical in architecture, optimization, and data, differing only in their attention masking strategy and corresponding language modeling objective during text pretraining (see Table 1). Specifically, the checkpoints include `enc`, a bidirectional encoder trained with masked language modeling (MLM); `dec`, a causal decoder trained with next-token prediction; and `dec-to-enc`, which starts as a causal decoder and is annealed at the end of training by removing the causal mask and switching to MLM. For the vision encoder, we employ the vision component of `siglip2-base-16b-512` (Tschannen et al., 2025), an 86M-parameter vision transformer contrastively trained on billions of image–text pairs. For simplicity, we refer to each ablation model using the name of its underlying text backbone (e.g., the vision retriever initialized from `enc` is denoted `enc`).

All ablations thus stem from iso-data controlled setups, and as further described, are further trained on the same data sequence, with the same batch sizes, optimizers, schedulers and on the same hardware.

**Model Architecture.** We design vision-language models following current standard training practices. In line with most recent work, we employ the early fusion architecture (Alayrac et al., 2022) illustrated in Figure 2, in which visual patch embeddings produced by the vision encoder are

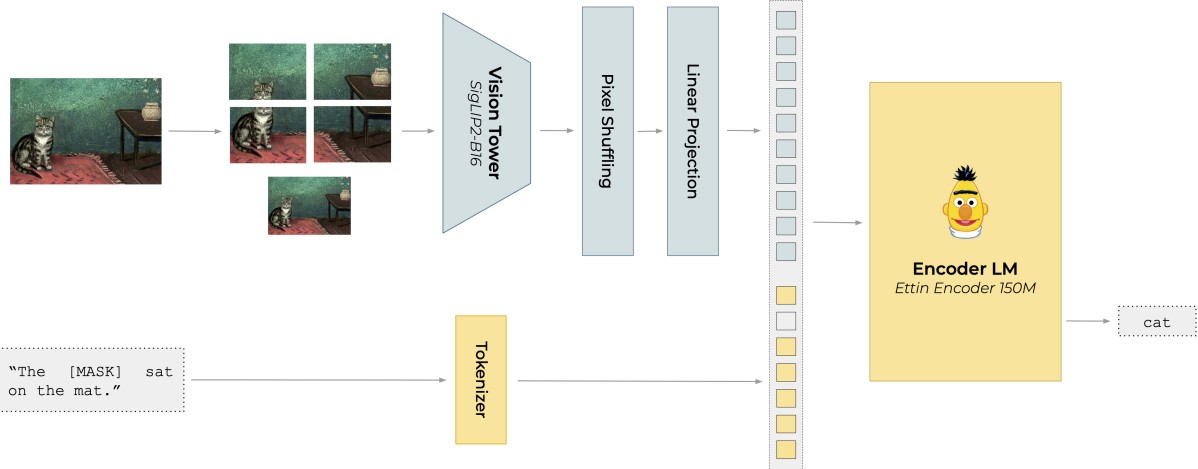

*Figure 2.* **MLM-based early fusion architecture.** The visual encoder produces patch representations, which are passed to a language model. Our end-to-end bidirectional attention fused architecture is trained with Masked Language Modeling objectives and is perfectly suited for sequence and token-level representation tasks.

projected into the language model input embedding space and concatenated with text token embeddings to encourage joint processing (Li et al., 2022; Alayrac et al., 2022; Wang et al., 2024; Yang et al., 2025; Marafioti et al., 2025). As described in Section 2.1, we generalize the training loss to function both with causal and masked language modeling objectives. To handle dynamic resolutions, we split large images into 512×512 pixel tiles as expected by the SigLIP encoder[1]. Following current standard practices, we further process a downscaled version of the full image to improve inter-tile consistency and global visual understanding (Lin et al., 2023; Ye et al., 2023). The vision encoder produces 1024 pixel patch representations for each tile[2], which we compress to 64 tokens through *pixel shuffling* (Shi et al., 2016) with a compression ratio $r = 4$, following prior work on models of comparable size (Marafioti et al., 2025). We highlight the impact of image resolution and this parameter on the number of visual tokens in Appendix C.7.1.

**Training Procedure.** Our experiments focus on retrieval performance. We employ a standard biphasic training procedure, in which we first run modality alignment to train a pretrained textual language model to understand visual inputs through language modeling objectives (Liu et al., 2023b) (Section 2.1), then rely on a second text-image contrastive learning phase to learn efficient image representations (Radford et al., 2021) (Section 2.2). We further describe the general setup, and detail specific modifications

to the default training procedure in the experiment section.

### 2.1. Modality Alignment

We align the vision encoder tower with the language model by training the image embedding projection layer to map visual features into the language model embedding space. The pretrained language model is also fine-tuned with Low-Rank Adapters (LoRA) (Hu et al., 2021), allowing both image and text models to adapt jointly while reducing the risk of monomodal performance collapse (Alayrac et al., 2022; Liu et al., 2023b; Laurençon et al., 2024c; McKinzie et al., 2024; Marafioti et al., 2025).

**Alignment Loss.** For decoder-based models, we train with Causal Language Modeling (CLM) loss on the text tokens, as standardly done in VLM modality alignment:

$$\mathcal{L}_{\text{CLM}} = -\sum_{t=1}^{T} \log P_\theta\big(x_t \mid x_{<t}\big), \quad (1)$$

where $x_{<t}$ denotes all tokens preceding position $t$. We generalize this training scheme to bidirectional encoders models, by using the Masked Language Modeling (MLM) loss on the textual tokens:

$$\mathcal{L}_{\text{MLM}} = -\sum_{t \in \mathcal{M}} \log P_\theta\big(x_t \mid x_{\setminus \mathcal{M}}\big), \quad (2)$$

where $\mathcal{M}$ is the set of masked token positions and $x_{\setminus \mathcal{M}}$ is the input with those tokens masked out.

**Modality Alignment Corpus.** Models are modality aligned on a large corpus in large parts derived from The Cauldron 2 (Laurençon et al., 2024c) and Docmatix (Laurençon et al., 2024a). Our objective being to

---

[1]Images are downscaled (or upscaled) so that the lengths and widths reach a multiple of 512 pixels to preserve the aspect ratio, padding is used on the smaller side when necessary (i.e. a 1024x1000 px image would be padded to 1024x1024 px).

[2]The SigLIP tower takes 512x512 px images and process them by 16x16 px patches (Dosovitskiy et al., 2020). This results in $(512/16)^2 = 1024$ patches.

train document focused retrieval models, we use an adjusted training mixture that upsamples images containing text and documents with varying level of complexities. Our final training corpus consists of approximately 2B text tokens, and includes diverse sources such as web pages, books, and scientific papers. Mixture details are given in Appendix A.3.1. We note that controlling the exact data distribution during this phase enables the models we train to specialize early and achieve good document focused downstream performances which many large models struggle with (Liu et al., 2023a).

**Parameters.** All models are trained using a masking ratio of 0.5 and user-prompt masking to avoid overfitting on chat-template format (Huerta-Enochian & Ko, 2024; Shi et al., 2024; Allal et al., 2025). We employ WSD scheduler (Hu et al., 2024b) with the first 5% of the training as warmup, the last 20% as decay and a maximum learning rate of 1e-4. The ablation models are aligned on 3.5B tokens. We provide additional details on the training setup in Appendix A.1.

## 2.2. Contrastive Post-Training

Once the language model has learned to process image tokens jointly with text tokens, we specialize models through a contrastive post-training stage designed to enhance the semantic representation of the output embeddings produced by the model (Reimers & Gurevych, 2019).

**Post-training Pairs.** The post-training dataset used as starting point in our ablations comprises 118k document-query pairs from the ColPali corpus (Faysse et al., 2025) as well as another 118k of natural image-description pairs from the MSCOCO train set (Lin et al., 2015).

**Contrastive Loss.** We employ the InfoNCE loss (van den Oord et al., 2019), defined as:

$$\mathcal{L}_{\text{InfoNCE}}(\mathbf{q}, \mathbf{d}^+) = -\log \frac{\Phi(\mathbf{q}, \mathbf{d}^+)}{\Phi(\mathbf{q}, \mathbf{d}^+) + \sum_{\mathbf{d}^- \in \mathcal{N}_q} \Phi(\mathbf{q}, \mathbf{d}^-)},$$
(3)

where $\mathbf{d}^+$ denotes the positive target for the query $\mathbf{q}$, $\mathcal{N}_{\mathbf{q}} = \mathcal{N}_{\mathbf{q}}^{\text{in}} \cup \mathcal{N}_{\mathbf{q}}^{\text{hard}}$ the set of negative targets (in-batch and hard negatives when mentioned), and $\Phi(\mathbf{q}, \mathbf{d})$ one of the similarity functions between the token(s) of the query and the documents.[3] For general-domain post-training we compute the loss symmetrically (Radford et al., 2021).

**Batches Curation.** In contrastive learning, batch diversity critically impacts retrieval entropy. Overly heterogeneous batches lead to trivial retrievals, while curated batches yield

richer training signals. We employ task-aware batching (Li et al., 2023), grouping documents by source to ensure a homogeneous batch composition.

## 2.3. Ablation Evaluation Setup

The contrastively trained models are evaluated on retrieval and zero-shot classification tasks across multiple domains. Although the main focus remains document retrieval capabilities, evaluated by aggregating scores from the ViDoRe and ViDoRe v2[4] (Macé et al., 2025) benchmarks (nDCG@5), we also assess more generalist image retrieval capabilities by selecting tasks from MIEB (Xiao et al., 2025a). For natural image retrieval, we aggregate MSCOCO retrieval (Lin et al., 2015) and Flickr30k retrieval (nDCG@10) (Plummer et al., 2016) test sets. Finally, following practices in (Muennighoff et al., 2022), we assess both zero-shot and fine-tuning abilities of our models on general classification tasks. Specifically, we measure classification accuracy by fine-tuning a logistic regression head on top of our model's embedding on Stanford Cars (Krause et al., 2013) and Food101 (Bossard et al., 2014), and we evaluate zero-shot performance on FER2013 (Khaireddin & Chen, 2021) and EuroSAT (Helber et al., 2019) and aggregate the results.

## 3. What Makes a Great Visual Retriever?

Vision-language retrievers built upon existing generative VLMs often inherit design choices and weights that may not be well suited for all embedding tasks. Here, we analyze these critical design choices hoping to derive clear insights for developing efficient visual retrievers. Importantly, although we assess design decisions at different stages of the training pipelines, evaluation are always done end-to-end on the final evaluation signal.

### 3.1. Modality Alignment Design

**Modality Alignment through LM improves document understanding.** According to benchmarks such as MIEB (Xiao et al., 2025a), dual encoder models explicitly trained on contrastive image-text tasks outperform repurposed VLMs in natural image classification tasks.

To assess this, we train an encoder and a decoder vision-language model using the methodology described in Section 2 on a mix of natural image and document data (in both alignment and contrastive training). We compare them with *SigLIP2-FT*, the 378M dual vision encoder model whose vision component is used by the vision encoder of both VLMs, and with the larger *SigLIP2-FT Large* (881M parameters). Both SigLIP-FT models are fine-tuned in

---

[3]We use the last (EOS) token for causal models, and mean pool all sequence tokens for bidirectional encoders for single-vector models. Alternatively, we use all document and query tokens without pooling for late interaction matching (Faysse et al., 2025). Details in Appendix A.2

[4]We report only the English splits of ViDoRe v2, as our base models are trained on English data only.

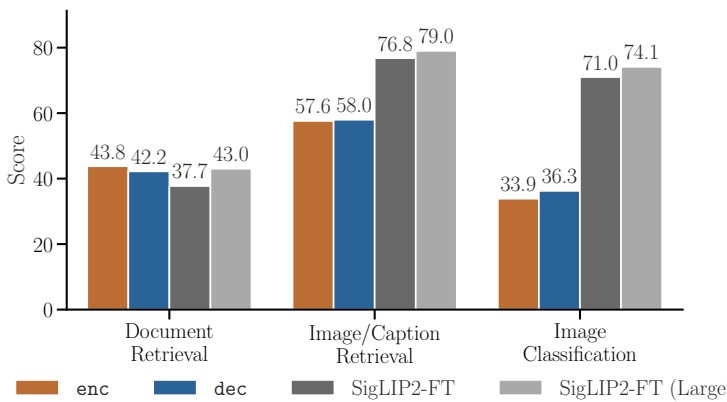

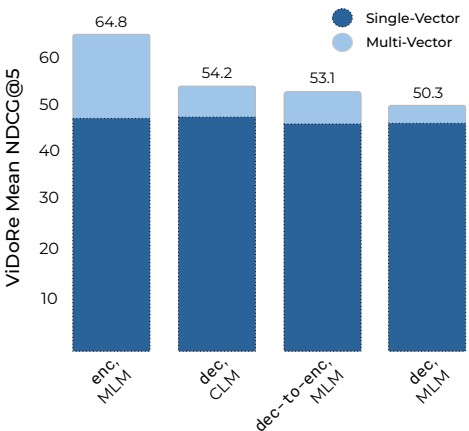

*Figure 3.* **Impact of Modality Alignment objective on downstream tasks.** Early Fusion of vision and text models boosts document retrieval tasks regardless of the LM objective, but degrades natural image and classification tasks w.r.t. the standalone *fine-tuned* vision model SigLIP. Reported scores are aggregated MIEB scores (nDCG, Accuracy.)

*Figure 4.* **Impact of attention masks and training objectives on document retrieval performances.** We report the average nDCG@5 on English splits of ViDoRe benchmarks for models post-trained on ColPali.

the same conditions as the VLMs, and initialized from pre-trained weights from scratch on billions of text-image pairs[5]. As shown in Figure 3, the two early fusion VLM variants severely underperform the SigLIP2-FT dual encoders on natural image tasks. In contrast, they achieve significant gains on document retrieval tasks (+6.1 nDCG@5 on ViDoRe and ViDoRe v2 datasets w.r.t. base), even edging out *SigLIP2-FT Large* that contains 3.5x vision parameters more than both VLMs. This confirms large-scale contrastive training remains best for high-level image representation tasks (natural images), but sequentially combining a vision model with a pretrained language model facilitates document representation tasks, even with significantly less contrastive post-training. As the rest of this paper shows, steering away from the dual encoder architecture further enables improving performance through many avenues other than text to image contrastive training, for which supervised training samples can be hard to obtain.

**Scaling the modality alignment phase for better token representations.** Prior work shows that scaling the modality alignment phase of VLMs improves their generative abilities (Beyer et al., 2024; McKinzie et al., 2024; Wang et al., 2024). We test whether similar gains hold in retrieval by contrastively fine-tuning enc checkpoints during MLM modality alignment. Although document retrieval improves consistently with more modality alignment data by largely surpassing the vision encoder evaluated in isolation and showing clear scaling benefits, natural image tasks plateau past 1B tokens, far from the standalone dual encoder baseline (Figure 7).

**Bidirectional attention fully unlocks Late Interaction.**

---

[5]We report the performance of the untrained *off-the-shelf* SigLIP in Appendix C.1

*Table 2.* **Effect of image resolution on VL encoder abilities.** Document retrieval performance increases with higher image resolution. Further annealing the encoder on high-resolution images (HR Cooldown) at the end of modality alignment yields additional gains. By contrast, for non-document tasks, raising the resolution tends to degrade performance.

| | HR Cooldown | Document Retrieval | Image/Caption Retrieval | Image Classif. | **Average** |
|---|---|---|---|---|---|
| 512px | ✗ | 30.7 | 58.8 | **41.4** | 43.6 |
| 1024px | ✗ | 42.2 | **58.9** | 37.2 | **46.1** |
| 2048px | ✗ | 43.8 | 57.6 | 33.9 | 45.1 |
| 2048px | ✓ | **45.8** | 57.8 | 33.7 | 45.8 |

Inspired by the effectiveness of bidirectional attention in text-only retrieval (Gisserot-Boukhlef et al., 2025; Weller et al., 2025)[6], we investigate if it surpasses causal attention in *visual document retrieval*, particularly when using the multi-vector late interaction matching common in SOTA visual retrievers (Khattab & Zaharia, 2020; Faysse et al., 2025). Figure 4 reports single vector and late interaction results on the ViDoRe benchmark for various model variants. On top of the standard enc (MLM) and dec (CLM) models, we evaluate the dec-enc and the dec models modality aligned with MLM objectives to determine whether bidirectional attention capabilities can be obtained in later stages of training. The models are contrastively post-trained using the training set of ViDoRe.

Single-vector embedding results are close between bidirectional and causal attention models for document retrieval. We observe no significant difference between enc and dec models.

---

[6]Chen et al. (2025) investigate post-hoc removal of the attention mask during visual retrieval fine-tuning.

*Table 3.* **Impact of contrastive training mixtures.** Incorporating text-only pairs improves document retrieval but degrades others. Synthetic image-caption pairs from *NatCap* enhance classification scores.

| | Document Retrieval | Image/Caption Retrieval | Image Classification | **Average** |
|---|---|---|---|---|
| Baseline CL Mix | 43.9 | **57.2** | 36.1 | 45.7 |
| + *Text→Text Pairs* | 45.6 | 53.2 | 35.7 | 44.8 |
| + *Synth. Img.→Caption Pairs* | **45.8** | 54.4 | **49.9** | **50.0** |

Intuitively however, bidirectional attention makes a huge difference when used in late interaction settings, substantially exceeding the causal counterpart by +10.6 nDCG@5.

Causal decoders are incapable of correctly contextualizing image or text token representations seen at the beginning of the sequences. *This is a key result as almost all current visual retrievers, including late interaction variants, are causal models, clearly indicating some performance is left on the table.*

Removing the causal attention mask during training does not suffice to recover the enc late interaction performance at these data regimes. This indicates converting trained decoders as late interaction retrievers is highly non trivial, and confirms the insights from Weller et al. (2025); when possible, training encoder models from scratch remain better for retrieval tasks.

### 3.2. Contrastive Training Design

The previous subsection established bidirectional encoder models to often be the best option when training visual retrievers. We now assess contrastive training choices and only report results for the encoder model for simplicity.

**Image resolution benefits are task-specific.** Image resolution plays a critical role in VLM generative capabilities, notably in document-focused tasks, as higher-resolution inputs enables the model to capture finer visual cues (Hu et al., 2024a; Marafioti et al., 2025). Modality alignment is done at a fixed image resolution of 1024 pixels (longer side) and we report scores of contrastive training runs with varying settings in Table 2. To vary the resolution, images of the highest quality available are scaled to the desired size (often downscaled) before being fed to the image tokenizer. Our findings confirm that embedding tasks are strongly sensitive to image-resolution. In particular, *training with higher resolution inputs substantially improves the results on visual document retrieval benchmarks*, consistent with prior work in generative settings (Beyer et al., 2024; McKinzie et al., 2024). Furthermore, adding a cool-down phase by increasing resolution towards the end of the modality alignment phase yields additional gains. This suggests that models can adapt their attention mechanisms to finer details when exposed to increased resolution. Interestingly, these findings do not hold in natural image tasks, where high resolution can even degrade performance.

**Increasing the pool of contrastive pairs.** A severe limitation that current visual retrievers face is the lack of large volumes of high quality (document image, query pairs). Previous work (Ma et al., 2024; Faysse et al., 2025; Jiang et al., 2025; Zhang et al., 2025a) has relied on a mix of re-purposed existing visual question answering datasets and synthetically generated queries with external LLMs. Even put together however, the field is only a year old, and these datasets remain small in size and often of poor quality.

A central question in our study is whether the abundance of *text-only* query–document pairs can be exploited to improve *visual* retrieval via cross-modal capability transfer. To probe this, we run contrastive training under three regimes. Unlike prior work that "warms up" visual retrievers or trains exclusively with text-only pairs (Ma et al., 2024; Jiang et al., 2024), we *interleave* text-only pairs and text–image pairs throughout training at a 1:1 ratio. The dataset sources are detailed in Appendix A.3.3

As reported in Table 3, incorporating text-only pairs yields a sizeable improvement on visual document retrieval (+1.7 NDCG@5), indicating clear cross-modal transfer, likely facilitated by the backbone's jointly learned text–image embedding space. This result suggests that domain-specific training corpora can be assembled irrespective of native modality, reducing duplication of effort and lowering data-collection costs.

We further evaluate training with *NatCap*, a corpus of natural images paired with synthetic, highly detailed captions (see Appendix A.3.2). This scaling step improves downstream performance across the board—most notably on natural-image tasks, and with a smaller but consistent gain on document retrieval (+0.2 NDCG@5). Together, these findings underscore the importance of scaling contrastive learning with high-quality data, but which doesn't need to be exclusively image document focused.

## 4. Building a Small Visual Retriever

### 4.1. Training

Putting together the results from our experiments, we devise a training recipe for a small visual document retriever *ModernVBERT*. It combines a state-of-the-art 150M

*Table 4.* **Performance on ViDoRe.** Our model *ColModernVBERT* offers the best performance-size tradeoff, significantly outperforming existing sub-1B models and matching the performance of models up to 10x larger with substantially lower inference CPU latency Details and GPU latencies in Appendix C.7.2. Models marked with * are not specifically trained for VDR. Bold values indicate the best performance amongst sub-1B models.

| | Late Interaction | Model Size (B) | ViDoRe (v1) | ViDoRe (v2,eng) | Average | Latency (ms) |
|---|---|---|---|---|---|---|
| *≥ 1B Parameters* | | | | | | |
| MoCa-3B (Chen et al., 2025) | | 3.75 | 80.1 | 53.8 | 66.9 | 158 |
| VLM2Vec (Jiang et al., 2025) | | 4.15 | 49.8 | 36.5 | 43.1 | 211 |
| GME-Qwen2 (Zhang et al., 2025a) | | 8.29 | 89.9 | 61.8 | 75.8 | 412 |
| E5-V (Jiang et al., 2024) | | 8.36 | 62.7 | 49.4 | 56.1 | 434 |
| ColPali (Faysse et al., 2025) | ✓ | 2.92 | 81.6 | 56.8 | 69.2 | 175 |
| ColQwen2.5 (Faysse et al., 2025) | ✓ | 3.75 | 89.5 | 61.5 | 75.5 | 158 |
| Jina-v4 (Günther et al., 2025) | ✓ | 3.75 | 90.4 | 60.1 | 75.2 | 158 |
| NemoRetriever-3B (Xu et al., 2025) | ✓ | 4.40 | 91.0 | 66.3 | 78.7 | 155 |
| *≤ 1B Parameters* | | | | | | |
| Jina CLIP* (Koukounas et al., 2024) | | 0.22 | 17.6 | 14.0 | 15.8 | **14** |
| BGE Visualized M3* (Zhou et al., 2024) | | 0.87 | 12.4 | 10.2 | 11.3 | 38 |
| SigLIP2-FT Large (Tschannen et al., 2025) | | 0.88 | 53.2 | 32.8 | 35.4 | 25 |
| ColFlor (Masry & Hoque, 2024) | ✓ | 0.17 | 68.8 | 43.0 | 55.9 | 17 |
| *BiModernVBERT* (ours) | | 0.25 | 63.6 | 35.7 | 49.7 | 20 |
| ***ColModernVBERT* (ours)** | ✓ | 0.25 | **81.2** | **56.0** | **68.6** | 20 |

text bidirectional encoder (Weller et al., 2025) with the ModernBERT architecture (Warner et al., 2024a) and a small vision encoder SigLIP2-16B-512 of 100M parameters (Tschannen et al., 2025).

**Modality Alignment.** We run the modality alignment phase for both models with a MLM objective on 10B tokens, 3 times longer than during our experiments[7]. To boost document understanding, we augment the input image resolution from 1024px to 2048px during a modality alignment cooldown stage (2B tokens). We call the resulting model *ModernVBERT*.

**VDR Specialization.** Following the findings of Section 3.2, we scale the contrastive training mix from previous experiments to combine document–query pairs with text-only pairs, and use 1 hard negatives for each document-query pair and 2 for each text-only pairs. We opt for a 2/1 text-to-image ratio following our ablation results detailed in Appendix C.4.1. This results in *ColModernVBERT*, a compact multi-vector retriever model. For reference, we also train *BiModernVBERT*, a single vector variant. More training details are provided in Appendix A.1.

**Evaluation.** We evaluate our approach on the ViDoRe benchmark suite (Faysse et al., 2025; Macé et al., 2025)[8]. We compare our approach against CLIP-like encoders (Radford et al., 2021; Zhai et al., 2023), as well as

recent multi-vector VDR models (Faysse et al., 2025; Meng et al., 2025; Chen et al., 2025). We present the baselines in detail in Appendix B.

### 4.2. Results

***ColModernVBERT*.** The resulting model, *ColModernVBERT* showcases strong performances on visual document retrieval benchmarks, especially relative to its size category (Figure 1). Despite having over 10 times less parameters than models such as ColPali released only a year ago, it is only 0.6 nDCG@5 points below on the aggregated ViDoRe benchmark scores (Table 4). It also edges many larger single-vector repurposed VLM models released within the year (Chen et al., 2025; Jiang et al., 2024; 2025). It however falls short of best models on ViDoRe which are built on larger decoder VLMs pretrained and aligned on billions of tokens of text and image data.

Most sub-1B parameter models evaluated on document retrieval benchmarks are dual encoder models, since early fusion generative models that perform well are not common at this scale. The most related model is a 176M late interaction model, ColFlor (Masry & Hoque, 2024), trained from the Florence2 model (Xiao et al., 2023). ColFlor is 12.7 nDCG@5 points under *ColModernVBERT*. *ColModernVBERT* also largely outperforms off-the-shelf dual encoders, even when those have substantially larger parameter counts. These results highlights the benefits of multiphase training and early fusion architectures for multimodal document related tasks, even at smaller parameter counts. We also attribute the strong performance of

---

[7]Appendix C.3 details the scaling dynamics that informed our choice.

[8]We only report the English splits as the base model is trained for English only.

*ColModernVBERT* at smaller model sizes to the symbiosis of native bidirectional attention and Late Interaction matching, which largely boosts performance relative to comparable decoder models (Section 3.1).

**Speed.** As noted by Xiao et al. (2025b), multi-vector visual retrievers are not bottlenecked in their inference speed by the late interaction matching operation, but rather by the latency required to encode queries with the text model. Our model demonstrates that strong performance is not incompatible with speed, even when running inference on consumer CPUs, which is the standard setting in most industrial local deployments of text embedding models. Latencies are computed by averaging query encoding times of all NanoBEIR queries, which are 23.4 word and 147.5 character long on average, and are run with batch size 1 to replicate online use cases. To prevent RAM bottlenecks, we benchmark on very high RAM (2TB) CPU cloud environments, but note models larger than 3B parameter require more than 12 GB RAM to run optimally.[9] (Table 4). *ModernVBERT* achieves more than a 7x speedup on CPU over models with similar performances on ViDoRe. We further report model latency results on GPU hardware in Appendix C.7.2. We notably demonstrate that with batched inference, *ModernVBERT* based query encoders are able to encode 5000 queries per second on Nvidia H100 GPUs. ModernVBert's small model size also enables efficient batching when encoding documents.

## 5. Related Work

**Repurposing VLMs for Representation Learning.** Motivated by the zero-shot performances of generative VLMs (Alayrac et al., 2022; Lucas Beyer* et al., 2024; Bai et al., 2023), recent studies have explored repurposing these for multimodal embedding tasks (Ma et al., 2024; Faysse et al., 2025; Jiang et al., 2025; Zhang et al., 2025a). As backbone generative models improved, retriever performance improved as well showcasing the central impact of language model pretraining and modality alignment (Xu et al., 2025; Nussbaum et al., 2025). These model remain inherently constrained by their causal attention mechanisms which has been shown in text settings to limits represational efficiency (Gisserot-Boukhlef et al., 2025; Weller et al., 2025). Recent work attempts to address this issue by modifying VLM attention during continual pretraining (Chen et al., 2025) or contrastive tuning (Jiang et al., 2025; Xu et al., 2025), but no recent work attempts to align natively bidirectional language encoder models with vision encoders. The recent release of long sequence text

encoders (Warner et al., 2024a; Boizard et al., 2025) makes this possible.

**Late Interaction in Visual Document Retrieval** To further boost performance, visual document retrievers leverage the late interaction mechanism (Khattab & Zaharia, 2020) which matches multiple query embeddings with multiple document embeddings through the MaxSim operation (Faysse et al., 2025; Günther et al., 2025; Xu et al., 2025). This enables more granular interactions between image and query tokens, at the cost of additional storage and a slight compute overhead during the matching operation. Efficiency gains have come from improving the storage costs through quantization (Bergum, 2025), token pruning (Faysse et al., 2024) and more recently the use of Matrioshka losses to compact multi-token representations (Xiao et al., 2025b). Ultimately, the performance bottleneck when running visual retrieval inference with such models now resides mostly in the necessity to rely on costly GPU hardware to encode queries, which sets apart text from vision retrieval. This paper fills this gap by using encoders that run on CPU, of parameter sizes comparable to commonly used local text embedding models (Chen et al., 2024; Enevoldsen et al., 2025).

## 6. Conclusion

In this paper, we systematically revisit design and training decisions of Visual Document Retrieval models. We uncover a tight synergy between bidirectional attention and the late-interaction multi-vector retrieval mechanisms standardly used in VDR, which highlights the fundamental limitation of repurposing decoder-style generative VLMs for retrieval. We analyze the impact of data regimes and image resolution downstream performance. To mitigate data scarcity in contrastive learning, we propose augmenting limited image-document/text-query pairs with larger, lower-cost corpora from other modalities. Guided by these insights, we train *ColModernVBERT*, a compact yet powerful 250M-parameter multimodal encoder that matches the performance of models up to 10 times larger on visual retrieval benchmarks. We release models and training code to help practitioners reduce cost and latency when deploying visual retrievers in real-world applications, and to encourage research on efficient multimodal embedding models.

**Future Work & Limitations.** By design, our analysis targets relatively small models. An important next step is to test whether the observed patterns persist at larger scales—for example, to more rigorously probe the interplay between late interaction and bidirectional attention. Our study focuses on English and while we expect the broad trends to generalize, it remains unclear how allocating parameters to additional languages trades off against the understanding of the vision modality, and to what extent

---

[9]With more standard CPU RAM settings such as those found in low-end servers or Google Colab (12GB RAM), models above 3B parameters must rely on memory offloading to run, which adds up to dozens of seconds of latency per query.

this penalizes English retrieval performance as the number of languages are scaled (Fernandes et al., 2023). Finally, although we center on retrieval and sequence-level zero-shot classification, the modality-aligned encoder can be fine-tuned for a range of token-level tasks, including OCR error detection, token-level classification, visual named entity recognition, visually grounded token-level object detection, contextual embeddings (Conti et al., 2025). We release our base model to encourage exploration of these directions.

## Impact Statement

**Environmental Costs.** Training *ColModernVBERT* required approximately 2,000 H100 GPU-hours in total, which we estimate corresponds to 41 kg of $CO_2$[10], based on standard assumptions of GPU power draw, datacenter efficiency, and grid carbon intensity. This estimate follows methodologies such as Green Algorithms (Lannelongue et al., 2021) and related analyses of the carbon footprint of machine learning (Strubell et al., 2019; Patterson et al., 2021). Across the entire project, all combined experiments totaled about 18k H100-hours. To mitigate costs and promote sustainable research practices, we release all model checkpoints and training artifacts to facilitate reuse, extension, and reproducibility without necessitating retraining. Additionally, this work shows efficiency gains with smaller models to aim to limit the inference costs of visual retrieval, and consequently reduce the environmental footprint. Our model performs query encoding efficiently on CPUs, keeping inference costs low and reducing barriers to adoption.

**Safety and Bias.** From a safety perspective, our encoder-only retriever poses less risk than generative models: it produces fixed-length embeddings rather than free-form content, reducing avenues for harmful content generation, hallucination, or deceptive outputs; nonetheless, retrieval systems can still propagate biases present in the underlying data, which we address through dataset curation open release.

## Reproducibility Statement

For transparency and to foster future work, we release our training data, model checkpoints (base models and adapters), and the complete codebase under the MIT License, as detailed in the main paper and repository. The supplementary material specifies training configurations for all models (also provided in the corresponding Hug-gingFace repositories), describes our synthetic data generation process, and reports expanded evaluation results to support exact replication.

## Detailed Contributions

**PT** is the first author of the project. Notably, he designed the modality alignement codebase, ran and supervised most large scale experiments across modality alignment and contrastive training, coordinated work, and was key in paper writing. **QM** ran large scale ablations on the data mixtures, contrastive training, and was responsible for the final training run. **MC** ran multiple experiments, including investigations into model merging and contributed to paper writing. **AL** focused on optimizing the data mixture and the contrastive training codebase. He was notably responsible for the initial cross-modality positive transfer results. **GV** and **PC** are senior contributors who helped with project framing, grant obtention, industry expertise and paper review. **MF** is last author and scientific lead of this work. He initiated and closely supervised the project from beginning to end, wrote the initial version of the multimodal contrastive training framework and greatly contributed to paper writing.

## Acknowledgments

This work was carried out within the framework of the LIAGORA "LabCom", a joint laboratory supported by the French National Research Agency (ANR) and established between ILLUIN Technology and the MICS laboratory of CentraleSupélec. This work was performed using HPC resources from IDRIS with grant AD011016393. We warmly thank Hippolyte Gisserot-Boukhlef and Nicolas Boizard for sharing the controlled experiments LM checkpoints, Antoine Chaffin for his feedback on the modality alignment codebase and insights on Ettin's modeling, as well as Andi Marafioti, Orr Zohar, and Miquel Farré for their valuable input and help on gathering the modality alignment dataset.

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

# A. Training

## A.1. Implementation and Resources

*Table 5.* Training details of our final models at each training stage. GPU Hours are on 80GB H100 GPUs.

| Model | Batch Size | Learning Rate | Training Steps | Training GPU Hours |
|---|---|---|---|---|
| **Modality Alignment** | | | | |
| *ModernVBERT*-base (Table 6) | 4096 | 1e-4 | 5500 | 1920h |
| **Contrastive Learning** | | | | |
| Generalist contrastive training (Table 8) | 256 | 2e-4 | 3917 | 80h |
| **Document Specialization** | | | | |
| Document-focused contrastive training w/ hard negatives (Table 8) | 64 | 2e-4 | 19602 | 160h |

We list hyperparameters and resource details in Table 5 for the various training stages of our final models. We employ ZeRO stage 1 optimizer (Rajbhandari et al., 2020) for our modality alignment runs. All ablation models are contrastively trained with gradient checkpointing (Chen et al., 2016) to reduce memory usage. All training runs are performed with FlashAttention 2.0 (Dao, 2023). For LoRA configurations, we consistently use a rank r of 32, lora_alpha of 32, and a dropout of 0.1. For the implementation, we start from m4[11] and ColPali[12] codebases for training, and use the MTEB[13] repository for evaluation.[14]

## A.2. Similarity Functions

**Single-Vector Similarity.** For single-vector models, we apply mean pooling for MLM-aligned encoders and end-of-sequence (EOS) pooling for CLM-based models and compute the cosine similarity of a query $q$ and a document $d$ as

$$\Phi_{\text{CosSim}}(\mathbf{q}, \mathbf{d}) = \exp(\cos(\mathbf{E}_q, \mathbf{E}_d)/\tau) \tag{4}$$

**Multi-Vector Similarity.** For multi-vector models, we adopt the standard late-interaction scoring function defined as:

$$\Phi_{\text{LI}}(q, d) = \sum_{i \in [\![1, N_q]\!]} \max_{j \in [\![1, N_d]\!]} \left\langle \mathbf{E}_q^{(i)}, \mathbf{E}_d^{(j)} \right\rangle, \tag{5}$$

where $\mathbf{E}_q^{(i)}$ and $\mathbf{E}_d^{(j)}$ denote token-level embeddings for the query and document, respectively.

## A.3. Data

### A.3.1. MODALITY ALIGNMENT MIXTURE

For our modality alignment trainings, we rely on The Cauldron dataset (Laurençon et al., 2024b) and its Docmatix extension (Laurençon et al., 2024a). Table 6 provides further details on the constitution of this dataset.

### A.3.2. *NatCap*

To enrich our contrastive learning data mixture, we construct *NatCap* (Natural Captions), a large-scale dataset containing around 333000 contextualized image–caption pairs. This dataset is created by generating synthetic captions, along with cross-class and in-class discriminative tags, from existing image classification datasets (see Table 7). For this purpose, we leverage Gemini-flash-2.5[15] which produces captions conditioned on both the image content and the accompanying dataset metadata, as illustrated in Figure 5. We detail the prompt below.

---

[11]SmolVLM trainer, https://github.com/huggingface/smollm
[12]https://github.com/illuin-tech/colpali
[13]https://github.com/embeddings-benchmark/mteb
[14]https://github.com/illuin-tech/modernvbert
[15]https://ai.google.dev/gemini-api/docs/models?#gemini-2.5-flash

*Table 6.* Aggregated statistics of modality alignment datasets from The Cauldron 2 (Laurençon et al., 2024c) and Docmatix (Laurençon et al., 2024a), showing image counts, QA pairs, token counts, and the proportional contribution of each subsection to the overall mixture.

| Dataset Subsection | # Images | # QA Pairs | # Tokens | % Mix |
|---|---|---|---|---|
| Captioning | 609,843 | 612,768 | 62,906,011 | 3.13 |
| Real-world VQA | 457,360 | 2,125,615 | 23,318,335 | 1.16 |
| OCR, Document Understanding | 2,499,258 | 11,415,478 | 426,806,479 | 21.21 |
| Chart/Figure Understanding | 539,743 | 24,444,120 | 30,315,784 | 1.51 |
| Table Understanding | 163,568 | 229,077 | 21,371,931 | 1.06 |
| Reasoning, Logic, Maths | 490,870 | 2,212,629 | 32,450,213 | 1.61 |
| Screenshot to Code | 547,974 | 548,296 | 336,299,551 | 16.71 |
| Text-only Instructions | 0 | 21,482,682 | 1,079,001,075 | 53.61 |
| **Total** | **5308616** | **63070665** | **2012469379** | **100.00** |

*Table 7.* **NatCap Dataset Composition.** *NatCap* spans 13 different sources covering various images types. The total dataset is composed of 333k pairs

| Dataset | Description | # Items |
|---|---|---|
| Caltech101 | General objects. | 3.000 |
| Caltech256 | General objects. | 30.000 |
| Cars | Car model classification. | 8.000 |
| Country211 | Country where the picture is taken. | 28.000 |
| DTD | Describable textures (texture attributes). | 4.000 |
| EuroSat | Land use / area zone type. | 16.000 |
| FER2013 | Facial emotion recognition. | 28.000 |
| FGCVAircraft | Aircraft model recognition. | 3.000 |
| Food101 | Food categories. | 75.000 |
| OxfordPets | Dog/cat species. | 3.000 |
| RESISC45 | Aerial scene / area zone type. | 18.000 |
| SUN397 | General scenes. | 109.000 |
| VOC2007 | General objects. | 8.000 |
| **TOTAL** | | **333000** |

### A.3.3. CONTRASTIVE TRAINING MIX

In this subsection, we describe the composition of our data mixes used in the contrastive training stages. Table 8 outlines the datasets included in each mix, including the Document-Focused variant employed for *ColModernVBERT*.

## B. Baselines Details

In this section, we describe the models evaluated in as comparison to our document retriever model.

**MoCa-3B** (Chen et al., 2025). A modality-aware continual pretraining model that transforms a causal vision-language model into a bidirectional multimodal embedding model, using interleaved image-text reconstruction and contrastive alignment to support cross-modal retrieval.

**GME-Qwen2** (Zhang et al., 2025a). A unified multimodal embedder built on Qwen2-VL (Wang et al., 2024), which produces shared embedding representations across text, image, and fused input modalities, enabling universal multimodal retrieval.

**VLM2Vec** (Jiang et al., 2025). A method that trains a vision-language encoder by converting a VLM through extensive contrastive post-training. Flagship model is based on the model Phi-3.5 (Abdin et al., 2024).

**E5-V** (Jiang et al., 2024). An adaptation of the E5 embedding approach to multimodal models: it trains only on text pairs yet bridges the modality gap to handle image inputs, reducing cost while achieving universal embeddings.

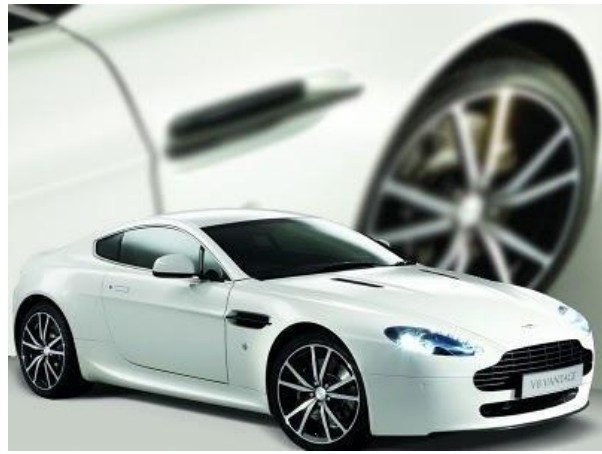

Class label: "Aston Martin V8 Vantage Coupe 2012"

*NatCap* caption: "A white 2012 Aston Martin V8 Vantage Coupe is showcased against a blurred background featuring a large car wheel."

*Figure 5.* Example from the NatCap dataset

*Table 8.* **Data mixes for contrastive trainings.** The *Generalist Mix* spans over 1M diverse pairs, while the *Document-Focused Mix* emphasizes document retrieval with extra ColPali epochs.

| Source | Description | Pairs | Epochs |
| --- | --- | --- | --- |
| **Generalist Mix** | | | |
| ColPali (Faysse et al., 2025) | Query–Document images for visual retrieval | 118k | 1 |
| MSCOCO (Lin et al., 2014) | Natural images with human-written captions | 118k | 1 |
| *NatCap (ours, subsampled)* | Diverse images with synthetic captions | 118k | 1 |
| RLHN (Thakur et al., 2025) | Text–text pairs for complex retrieval | 680k | 1 |
| **TOTAL** | | **1030k** | |
| **Document-Focused Mix** | | | |
| ColPali (Faysse et al., 2025) | Query–Document images for visual retrieval | 118k | 3 |
| RLHN (Thakur et al., 2025) | Text–text pairs for complex retrieval | 300k | 3 |
| **TOTAL** | | **1254k** | |

**ColPali** (Faysse et al., 2025). A vision-based document retrieval model that processes document pages as images (no OCR) and produces multi-vector embeddings via a late-interaction mechanism over PaliGemma (Beyer et al., 2024), enabling efficient and accurate retrieval.

**ColQwen2.5** (Faysse et al., 2025). An extension of ColPali (Faysse et al., 2025) using Qwen2-VL (Wang et al., 2024) as the backbone, carrying forward the late interaction retrieval paradigm over page image embeddings, capturing layout and textual context without OCR.

**Jina-v4** (Günther et al., 2025). A multimodal embedding model combining visual and textual inputs with support for multi-vector (late interaction) embeddings, using adapters over a unified backbone to excel on visually rich document retrieval.

**NemoRetriever** (Xu et al., 2025). An LI retriever that combines vision-language embeddings with a ColEmbed design, enabling high performance on visual document retrieval with structured patch matching and efficient similarity.

**Jina CLIP** (Koukounas et al., 2024). A smaller scale vision-language model using CLIP embeddings, applied to document retrieval tasks; although not LI, it offers a lightweight multimodal baseline.

**BGE Visualized M3** (Zhou et al., 2024). A vision-enhanced version of BGE M3 (Chen et al., 2024) that supports visual inputs and extends embedding models into multimodal domains.

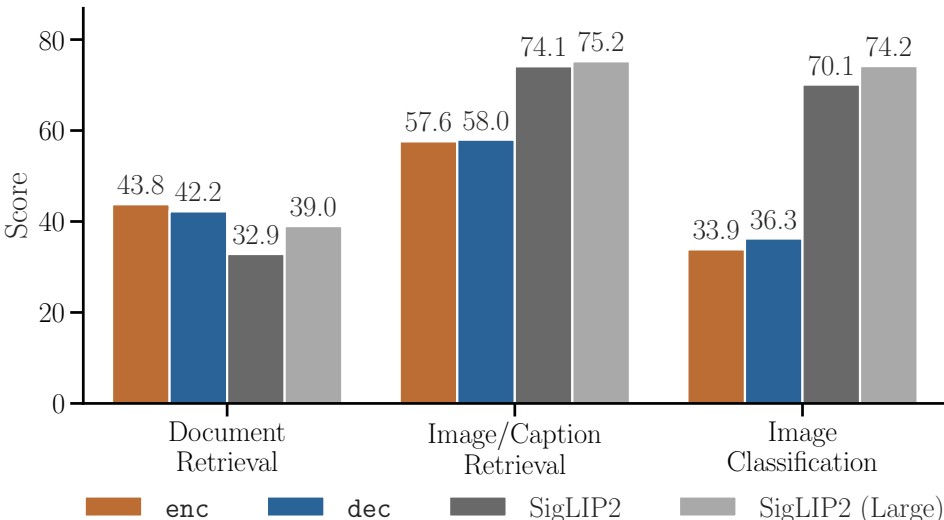

*Figure 6.* **Impact of Modality Alignment objective on downstream tasks.** Early Fusion of vision and text models boosts document retrieval tasks regardless of the LM objective, but degrades natural image and classification tasks w.r.t. the standalone *off-the-shelf* vision model SigLIP. Reported scores are aggregated MIEB scores (nDCG, Accuracy.)

**SigLIP2-L-512/16** (Tschannen et al., 2025). A multilingual vision-language bi-encoder model, which combines image and text modalities to yield unified embeddings across languages. This configuration handles images of 512x512 pixels and create subpatches of 16x16 pixels.

**ColFlor** (Masry & Hoque, 2024). A lightweight OCR-free visual document retriever with only 174M parameters built over Florence-2 and DaViT, delivering strong performance near ColPali with much lower computational cost and much faster encoding.

## C. Additional Ablations

### C.1. Performance Against Off-the-Shelf Dual Encoder

We study whether using *off-the-shelf* performances of the standalone vision encoder are not outweighing the burden of adding language parameters and re-training through language modeling, as proposed in our work. Figure 6 shows the results of the various models on the tasks described in Section 2. Similarly to Section 3.1, we observe that the early fusion model trained with LM objective significantly outperform the standalone vision encoder on document retrieval tasks (+10.9 nDCG@5). It even surpass the larger dual encoder (+4.8 nDCG@5) on these latest tasks. We note that the standalone vision encoder largely outperform the early fusion models on the other natural images tasks, supporting for the use of the SigLIP model for these tasks as found in various general benchmarks (Xiao et al., 2025a).

### C.2. Scaling the modality alignment phase for better token representations.

Prior work shows that scaling the modality alignment phase of VLMs improves their generative abilities (Beyer et al., 2024; McKinzie et al., 2024; Wang et al., 2024). We test whether similar gains hold in retrieval by contrastively fine-tuning `enc` checkpoints during MLM modality alignment. Figure 7 illustrates the results of post-trained checkpoints on diverse tasks. Although document retrieval improves consistently with more modality alignment data – largely surpassing the vision encoder evaluated in isolation and showing clear scaling benefits – natural image tasks plateau past 1B tokens, far from the standalone dual encoder baseline. This shows that document and natural image retrieval leverage different mechanisms and should not be optimized the same way. *Document Retrieval benefits from learning fine-grained interactions between image and text tokens through the language model, while the LM has limited utility for high level natural image tasks.*

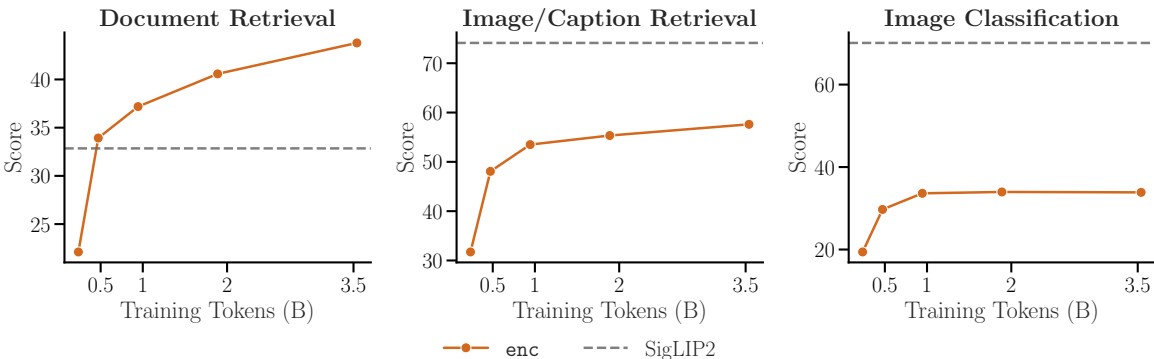

*Figure 7.* **Modality alignment scaling of early fusion encoders for up to 1 epoch (3.5B tokens) of data.** The dashed line indicates the vision encoder evaluated standalone without further training. Our findings show that retrieval tasks benefit from an extended modality alignment phase, particularly in document retrieval, where performance quickly surpasses that of the standalone vision encoder.

## C.3. Scaling Dynamics of Attention Masks

We compare the `enc` (MLM) approach with a traditional `dec` (CLM) objective. Figure 8 presents the performance of the two training objectives across a diverse set of tasks. For both objectives, document retrieval improves consistently with more modality alignment data, largely surpassing the vision encoder evaluated in isolation and showing clear scaling benefits. Comparing the two alignment approaches, we observe that while `dec` offers an advantage in low-data regimes, `enc` seems to catches up. In document retrieval tasks, it eventually surpasses `dec` and scales better.

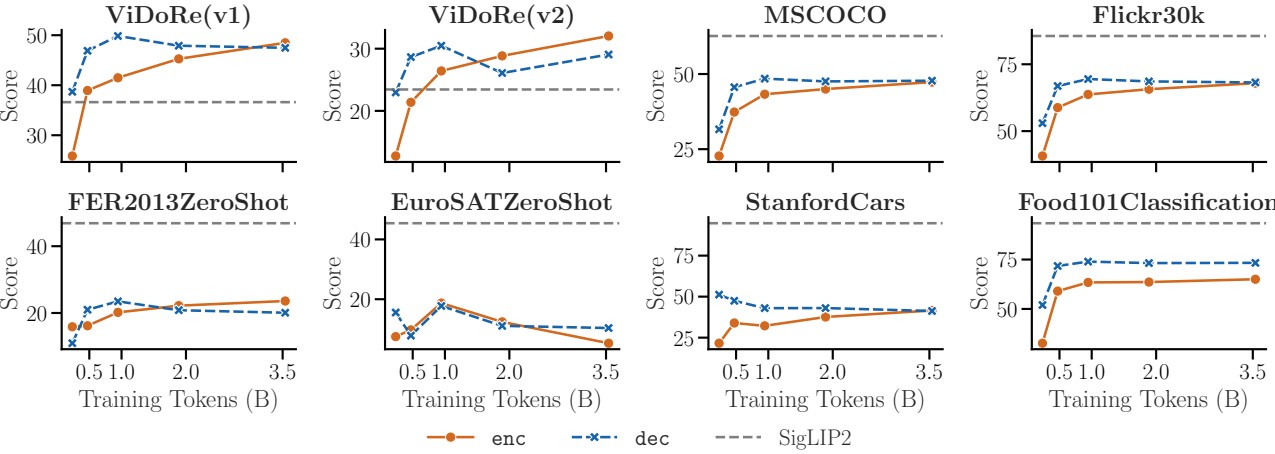

*Figure 8.* **Attention masks impact on modality alignment phase scaling.** The dashed line marks the vision encoder baseline. The orange curve shows the model initialized from a decoder LM with a *CLM* objective, and the blue curve shows the model trained with an *MLM* objective from an encoder LM. CLM performs better in low-data regimes, but MLM scales more effectively, surpassing CLM in document retrieval, while captioning and classification remain below the CLIP baseline.

## C.4. Bridging the Gap with Longer Contrastive Training

We study the impact of additional in-distribution training pairs on embedding tasks by scaling the contrastive training stage. Starting from the final checkpoint of our encoder-based ablation model, we double the contrastive dataset size at each step and train until convergence[16]. This setup tests whether scaling continues to improve performance. Figure 9 shows the scaling behavior. Performance improves overall with more in-distribution data. The vision-tower baseline is quickly surpassed on visual document benchmarks, and scaling narrows the gap on other tasks[17]. We note a plateau in captioning

---

[16]To avoid overfitting, we set an early stopping on an eval set. We limit the number of step to one epoch on the full dataset.

[17]Note that the models probably won't fully recover baseline vision-tower performance. This highlights the need to choose models according to use case (e.g., lightweight CLIP-like models for image classification).

and classification, pointing to the need for more diverse data.

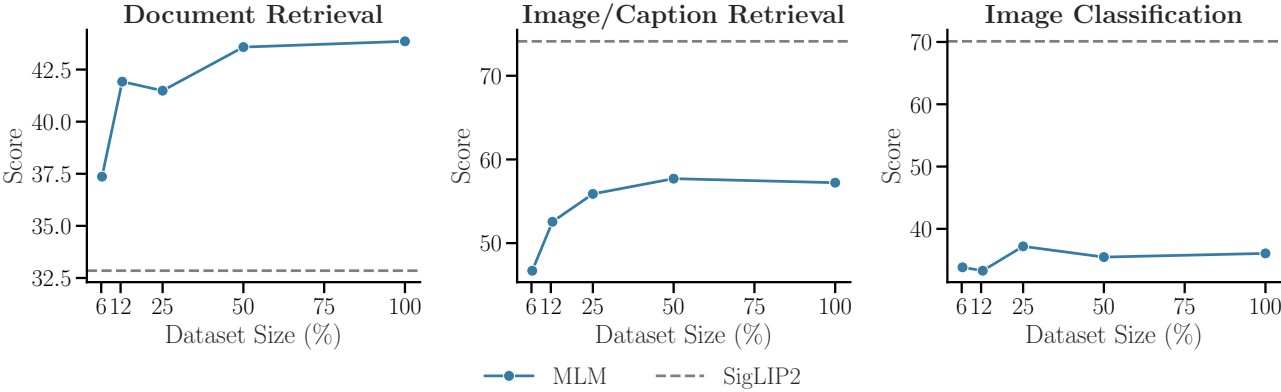

*Figure 9.* Contrastive training scaling. Each dot on the blue curve represents one fraction of the baseline contrastive training mix (ColPali + MSCOCO). Performance improves with more in-distribution data, surpassing the baseline on document benchmarks and narrowing the gap on image captioning. There is no clear improvement in image classification, highlighting the need for more diverse pairs.

### C.4.1. OPTIMAL TEXT-TO-IMAGE RATIO FOR DOCUMENT RETRIEVAL

Our findings in subsection 3.2 indicate that incorporating additional text-only pairs boosts document retrieval performance. While our initial experiment employed a 1:1 text-to-image ratio, we further investigate how varying this ratio impacts our broad set of tasks. We start from the best contrastive mix in Table 3, and vary the text-to-image ratio. As shown in Figure 10, increasing the number of text-only pairs *for a fixed amount of image pairs* consistently enhances retrieval performance. However, for natural image classification tasks, adding more text does not appear to provide benefits.

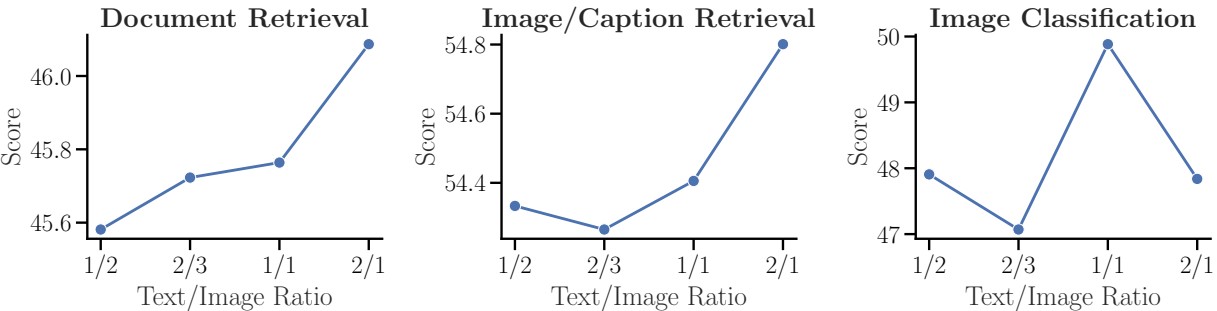

*Figure 10.* **Optimal text-to-image ratio in contrastive training mix.** Increasing the ratio in retrieval tasks consistently improves the performances.

## C.5. Late Interaction for Non-Documental Retrieval

We want to study if the multi-vector gains transfer to non-documental retrieval. To do so, we contrastively post-train our base model on our generalist post-training mix presented in Table 8. The late interaction generalist exhibits superior performance in retrieval setting, improving its single-vector performance by +20.2% (11.5 points), matching the performance of substantially larger VLM-based retrievers like E5-V (8.3B parameters, 67.5 points) and surpassing dual encoders like SigLIP (882M parameters, 56.7 points). This matches the capabilities observed in Section 3.1 for documental settings for models with native bidirectional attention, extending it to natural image tasks. This result extends the prevailing understanding from the document retrieval community, where the superiority of late-interaction is well-documented ((Khattab & Zaharia, 2020), (Chaffin, 2025), (Faysse et al., 2025)). While this performance gap is widely accepted for document retrieval, its applicability to caption matching tasks has not really been addressed. Our findings provide strong evidence that the fine-grained matching capabilities of late-interaction models are a key driver of performance in this domain too.

*Table 9.* **Generalist retrieval performances.** Late interaction benefits extend to non-documental retrieval tasks. Our multi-vector model increases its single-vector counterpart across all tasks, surpassing larger VLM-based retrievers.

| | Model Size | Document Retrieval | | Image/Caption Retrieval | | Average |
| | | ViDoRe(v1) | ViDoRe(v2) | MSCOCO (T→I) | Flickr30k (T→I) | |
|---|---|---|---|---|---|---|
| *CLIP Encoders* | | | | | | |
| siglip2-base-patch16-512 | 376M | 36.6 | 23.4 | 66.2 | 86.9 | 53.3 |
| siglip2-large-patch16-512 | 882M | 43.8 | 27.0 | 67.1 | 88.9 | 56.7 |
| clip-vit-base-patch16 | 151M | 25.5 | 20.4 | 50.3 | 76.8 | 43.3 |
| clip-vit-large-patch14 | 428M | 38.0 | 28.6 | 52.7 | 79.3 | 49.6 |
| *VLM-based Encoders* | | | | | | |
| VLM2Vec-Full | 4150M | 49.8 | 36.5 | 59.5 | 81.8 | 56.9 |
| e5-v | 8360M | 62.7 | 49.4 | 68.1 | 89.8 | 67.5 |
| *Early Fusion Encoders* | | | | | | |
| bge-visualized-base | 196M | 10.3 | 9.0 | 50.0 | 74.1 | 35.9 |
| bge-visualized-m3 | 873M | 12.4 | 10.2 | 39.6 | 69.0 | 32.8 |
| ***ModernVBERT*-embed** | 252M | 58.4 | 36.9 | 56.5 | 76.0 | 56.9 |
| ***ModernVBERT*-embed (multi-vector)** | 252M | 76.5 | 53.9 | 61.8 | 81.4 | **68.4** |

### C.5.1. MODEL MERGING

Our contrastive learning stage provides direct performance trade-offs on different tasks. Following recent trends, we evaluate how model merging techniques allow to mitigate performance degradation on specific tasks, while maintaining the performance enabled by the contrastive training (Sung et al., 2023; Dziadzio et al., 2024; Li et al., 2024; Zhang et al., 2025b). We merge our ablation model after modality alignment with the checkpoint after the full contrastive learning with two methods: SLERP (Ilharco et al., 2022) and average merging (Shoemake, 1985). For SLERP, we compare three values for the $\lambda$ coefficient (0.25, 0.5, 0.75). Section 11 displays the the trends with the best method (SLERP, $\lambda = 0.75$). As we can see, the merged model mitigates the performance drop in Image/Caption Retrieval tasks, while maintaining significant gains on Image Classification tasks. However, merging strongly degrades performance on Document Retrieval, showing that benefits of merging embedding models are task-dependent.

### C.5.2. CURRICULUM FOR DOCUMENT RETRIEVER CONTRASTIVE POST-TRAINING

*Table 10.* **Performance of *ModernVBERT* Document Specialisation Curriculums.** This table presents the performance of various contrastive training curriculums starting from *ModernVBERT*-base, on the ViDoRe(v1) and ViDoRe(v2) benchmarks. The generalist contrastive learning mix used in the last two models is detailed in Table 8. We see that a preliminary stage of generalist contrastive learning harms the final document retrieval performance, regardless of whether a multi-vector approach is used.

| | ViDoRe(v1) | ViDoRe(v2) | Average |
|---|---|---|---|
| *Document retrieval contrastive training starting checkpoint* | | | |
| *ModernVBERT*-base | **81.2** | **56.0** | **68.6** |
| + multi-vector generalist CL | 80.7 | 55.4 | 68.1 |
| + single-vector generalist CL | 80.6 | 54.0 | 67.3 |

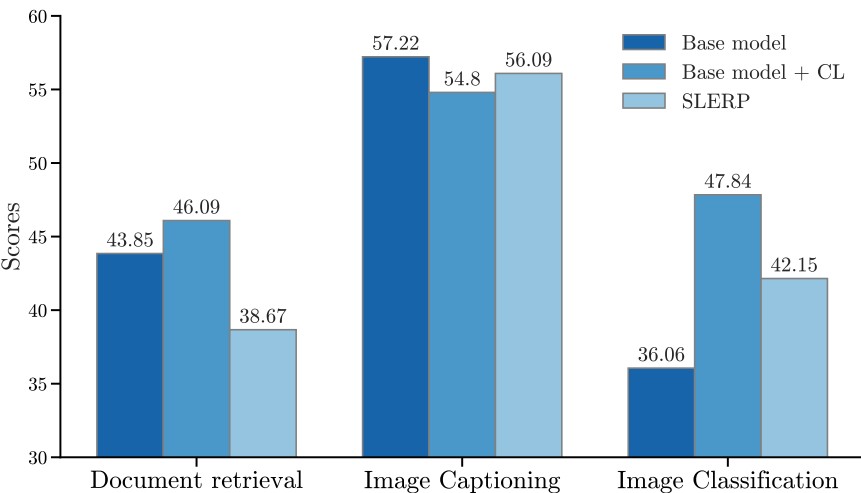

*Figure 11.* Merging model results across tasks. Benefits are task-dependent, with performance degradation w.r.t. both original models in Document Retrieval.

We conduct an ablation study to determine the optimal contrastive training curriculum for specializing *ModernVBERT* in document retrieval. Specifically, we investigate whether a preliminary generalist contrastive training phase, intended to leverage a larger dataset, improves downstream performance. As shown in Table 10, our results demonstrate that this initial generalist phase is detrimental to final performance ($-0.5\%$). The optimal strategy is to specialize the model on the target task directly after its initial Masked Language Modeling (MLM) alignment.

### C.6. Text-Only Retrieval

*Table 11.* Average NDCG@5 of *ColModernVBERT* on NanoBEIR, a text retrieval benchmark with multiple sub domains. *For multi-modal models, we only consider parameters of the text encoder

| Model | Params (M) | NDCG@5 |
|---|---|---|
| **Statistical** | | |
| BM25s | — | 0.559 |
| **Single Vector** | | |
| Jina Embeddings v4 | 3577* | 0.623 |
| E5-large-v2 | 335 | 0.605 |
| bge-m3 (Bi Encoder) | 567 | 0.590 |
| Qwen3-Embedding-0.6B | 600 | 0.567 |
| **Multi Vector** | | |
| LightOn GTE-ModernColBERT v1 | 149 | 0.669 |
| Jina ColBERT v2 | 137 | 0.642 |
| bge-m3 (Late Interaction) | 567 | 0.606 |
| ColBERT v2 | 110 | 0.593 |
| Colqwen2-v1.0 | 1580* | 0.593 |
| *ColModernVBERT* | 150* | 0.589 |
| Colqwen2.5-v0.2 | 3145* | 0.589 |

The results in Table 11 detail the performance of *ColModernVBERT* and other baselines on the NanoBEIR text retrieval benchmark. It achieves an average NDCG@5 score competitive with single and multi vector models specialized for text, even without explicit optimization for this modality. This performance is encouraging and indicates a promising direction for training a unified model for both text and image retrieval.

## C.7. Model Latency

### C.7.1. IMAGE RESOLUTION TRADEOFFS

Figure 12 presents the pixel shuffling trade-off. Processing larger images creates more visual tokens, leading to very long sequences (around $17'500$ tokens for a 2048x2048 px image with no pixel shuffling). Pixel shuffling allow to compress these sequence by concatenating the embeddings of spatially close patches. This diminishes the number of tokens for longer visual token embeddings. Table 12 presents the latency to process one image of various resolutions on one L4 GPU and CPU.

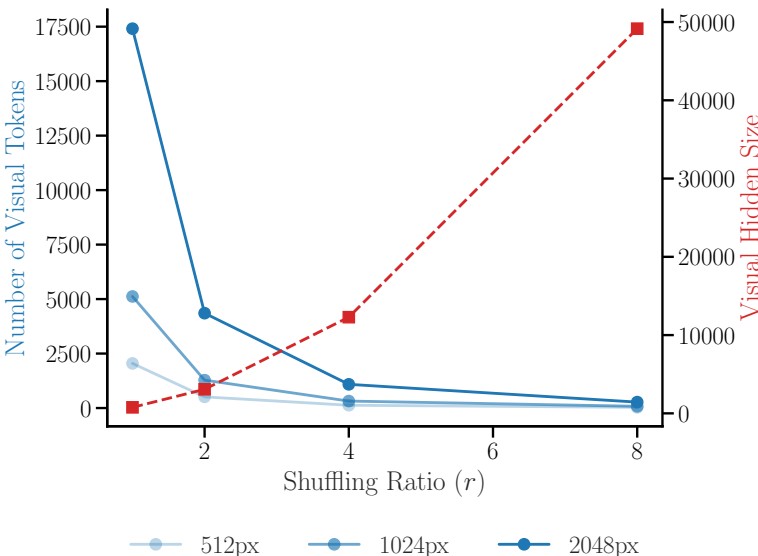

*Figure 12.* **Image processing parameters impact on visual tokens.** Here we assume a square image for simplicity. Scaling the image size introduces naturally more tokens, but having a large enough pixel shuffling ratio ($r \geq 4$) allows to counterbalance by concatenating spatially close patch representations.

*Table 12.* ***ModernVBERT*** **image processing latency**. Computing the average time to process a single image on GPU and CPU. The average is computed on 100 images. The values represent the mean latency in milliseconds, with the standard deviation included in parenthesis.

|        | Num. Visual Tokens | CPU Latency (ms) | GPU Latency (ms) |
|--------|--------------------|------------------|------------------|
| 512px  | 128                | $287.2_{(\pm7.8)}$   | $43.6_{(\pm1.4)}$   |
| 1024px | 320                | $1015.8_{(\pm58.1)}$ | $150.3_{(\pm2.5)}$  |
| 2048px | 1088               | $2572.0_{(\pm63.9)}$ | $363.4_{(\pm4.6)}$  |

### C.7.2. ONLINE QUERY ENCODING LATENCY

We evaluate the query embedding speed of our model on GPU. We use a single Nvidia H100 with 80GB of VRAM. As for Section 4.2, latencies are computed in batch size 1 to simulate online situations, and are averaged over all NanoBEIR queries. Only the text parameters are loaded and run, to minimize memory usage. Parameters are cast to bfloat16 and Flash Attention 2 is used. The resulting speeds are often much faster than those obtained by running inference through each model's reference implementation. Results are shown in Table 13). Interestingly in this setup where memory is not a bottleneck, model depth seems to be a large performance driver, sometimes more the parameter count. We finally evaluate batched GPU throughput. We use batches of size 512 by default and iteratively half it when memory is insufficient. We observe that *ModernVBERT* based models are extremely fast and can process 5000 queries per second. In the table, the reported figures correspond to the inverted throughput (latency per batch divided by the number of queries per batch). These speed and throughput gains are made possible due to a combination of size, and efficient hardware-informed design as well as the support of flash attention and sequence packing other models of the size often lack (Warner et al., 2024b).

*Table 13.* **Text query encoding latency.** The latency is computed both on high-end CPUs (1TB RAM, 128 cores) and GPU (Nvidia H100, 80GB) (mean ± std). Since only 649 queries are used, standard deviations are not reported in GPU batching mode (batches of 512 queries by default), for which we report the inverse throughput (average latency per batch divided by the batch size).

| | Late Interaction | Model Size (B) | CPU Latency (ms) | GPU Latency (ms) | GPU Batching (ms) |
|---|---|---|---|---|---|
| *≥ 1B Parameters* | | | | | |
| MoCa-3B | | 3.75 | $158_{(\pm147)}$ | $26_{(\pm3)}$ | 4.54 |
| VLM2Vec | | 4.15 | $211_{(\pm253)}$ | $21_{(\pm3)}$ | 2.82 |
| GME-Qwen2-7B | | 8.29 | $412_{(\pm411)}$ | $25_{(\pm1)}$ | 9.07 |
| E5-V | | 8.36 | $434_{(\pm379)}$ | $22_{(\pm2)}$ | 9.55 |
| ColPali | ✓ | 2.92 | $175_{(\pm113)}$ | $14_{(\pm1)}$ | 3.07 |
| ColQwen2.5 | ✓ | 3.75 | $158_{(\pm147)}$ | $26_{(\pm2)}$ | 26 |
| Jina-v4 | ✓ | 3.75 | $158_{(\pm147)}$ | $26_{(\pm2)}$ | 4.54 |
| NemoRetriever-3B | ✓ | 4.40 | $155_{(\pm118)}$ | $20_{(\pm2)}$ | 4.59 |
| *≤ 1B Parameters* | | | | | |
| Jina CLIP | | .22 | $14_{(\pm7)}$ | $6_{(\pm2)}$ | .69 |
| BGE Visualized M3 | | .87 | $38_{(\pm42)}$ | $10_{(\pm2)}$ | .77 |
| SigLIP2-L-512/16 | | .88 | $25_{(\pm8)}$ | $6_{(\pm1)}$ | .10 |
| ColFlor | ✓ | .17 | $17_{(\pm9)}$ | $8_{(\pm2)}$ | .31 |
| *BiModernVBERT* (ours) | | .25 | $20_{(\pm11)}$ | $14_{(\pm2)}$ | .20 |
| **ColModernVBERT** (ours) | ✓ | .25 | $20_{(\pm11)}$ | $14_{(\pm2)}$ | .20 |

## C.8. Per-Domain Performance Breakdown on ViDoRe

We report the full per-domain performance breakdown on both ViDoRe v1 and ViDoRe v2 in Tables 14 and 15, respectively.

*Table 14.* Per-domain nDCG@5 on ViDoRe v1. All models use Late Interaction. [†]TabFQuAD and Shift Project are French-language splits. **Bold** values indicate the best score per column.

| Model | Size | DocVQA | InfoVQA | TAT-DQA | arXivQA | TabFQuAD[†] | Energy | Government | Healthcare | AI | Shift Project[†] | Avg |
|---|---|---|---|---|---|---|---|---|---|---|---|---|
| *ColModernVBERT* | 0.25B | 55.7 | **86.4** | **73.2** | 75.0 | 76.1 | **93.7** | **93.3** | **96.4** | **97.5** | 64.9 | 81.2 |
| ColPali | 2.92B | **59.1** | 82.2 | 66.2 | **80.0** | **81.9** | 92.1 | 92.8 | 94.8 | 96.9 | **70.2** | **81.6** |

*ColModernVBERT* matches ColPali on average (81.2 vs. 81.6) while being 11× smaller, and outperforms it on 6 out of 8 English sub-tasks, most notably on TAT-DQA (+7.0) and InfoVQA (+4.2). The only meaningful gaps arise on the two French-language sub-tasks—TabFQuAD (−5.8) and Shift Project (−5.3)—which is expected given that *ColModernVBERT* relies on an English-only text backbone. No consistent weakness emerges across English document types, indicating that the performance advantages of *ColModernVBERT* generalise across domains.

On ViDoRe v2, *ColModernVBERT* remains within approximately 2 nDCG@5 points of ColPali across all sub-tasks, despite being 11× smaller. The gap is not concentrated in any particular domain, confirming that no systematic domain-specific weakness emerges beyond the expected degradation on non-English content. As discussed in Appendix C.9, replacing the text backbone with a multilingual encoder directly addresses the cross-lingual limitation.

## C.9. Multilingual Capabilities and Language Backbone

We evaluate our model on the multilingual splits of the ViDoRe v2 benchmark.

*Table 15.* Per-domain nDCG@5 on ViDoRe v2. All models use Late Interaction. **Bold** values indicate the best score per column.

| Model | Size | Biomedical Lectures | ESG Reports | Economics Reports | Avg |
|---|---|---|---|---|---|
| *ColModernVBERT* | 0.25B | 55.3 | 53.3 | **60.4** | 56.0 |
| ColPali | 2.92B | **57.1** | **55.4** | 57.7 | **56.8** |

*Table 16.* NDCG@5 on the multilingual splits of ViDoRe v2. *ColEuroVBERT* uses EuroBERT as its text backbone in place of the English-only encoder used by *ColModernVBERT*. Best values are highlighted in bold.

| Model | English | French | Spanish | German |
|---|---|---|---|---|
| *ColModernVBERT* | **0.560** | 0.376 | 0.277 | 0.225 |
| *ColEuroVBERT* | 0.483 | **0.389** | **0.425** | **0.342** |

To investigate whether the architectural benefits we identified (e.g., the synergy between bidirectional attention and multi-vector matching) generalize across languages, we trained a parallel variant, *ColEuroVBERT*, by swapping the English-only text backbone for a natively multilingual one (EuroBERT (Boizard et al., 2025)). The only change between the two models is the text encoder backbone; all other architectural choices remain identical.

Table 16 reports NDCG@5 scores on the English, French, Spanish, and German splits of ViDoRe v2. *ColModernVBERT* – which relies on an English-only base text encoder – shows a natural performance degradation on non-English retrieval tasks. Replacing the text backbone with EuroBERT significantly improves performance across all non-English languages and drastically reduces cross-lingual variance, at the cost of a moderate drop in peak English performance. These results suggest that the architectural design choices central to our approach are not language-bound, and that extending the framework to new languages is achievable either by adopting a multilingual text backbone or through techniques such as cross-lingual distillation.

---

### *NatCap* Annotation Prompt

You are an image annotator expert.

You will receive an image along with its classification label and the classification task scope, and your task is to provide contextualized metadata about it.

The output should be a JSON object with the following metadata fields:

- **caption**: A descriptive caption of the image accounting for its label. This should be a **unique** and concise sentence that describes the image in detail.

- **class_tags**: A list of tags that represents the image and can help identify the class. (e.g., for a car image with its model as a class, this could be some specific attribute of the car)

- **other_tags**: A list of tags that represents the image but can help identify the image among others of the same class. (e.g., for a car image with its model as a class, this could be its color or the background of the image)

- **is_image_class_explicit**: Boolean, could the class be inferred from the image alone? (e.g., the class is a country and you cannot necessarily infer it from the image alone, so this would be `false`)

Please ensure that the output is in valid JSON format.

**Example:**
You receive an image of what is clearly a car with its model as a class (here Audi TTS coupe 2012) for a car model classification task.
The output could be a JSON object like this:

```
{
  "caption": "A red Audi TTS coupe 2012 car parked on a sunny street
              in front of a sport shop.",
  "class_tags": ["sport coupe","four door coupe","17'' alloy wheels"],
  "other_tags": ["sunny street","parked","red","sport shop"],
  "is_image_class_explicit": true
}
```

———

Classification scope: {task_info}
Image label: {label}
Answer:

