# OpenReview forum: "ModernVBERT: Towards Smaller Visual Document Retrievers"
_ICML.cc/2026/Conference — ICML 2026 regular_

### Official Review · Reviewer_LN7s · 2026-03-10

**Soundness:** 4
**Presentation:** 4
**Significance:** 3
**Originality:** 2
**Overall Recommendation:** 4
**Confidence:** 4

**Summary:**

This paper systematically studies the design decisions for Visual Document Retrieval (VDR) models, challenging the prevailing approach of repurposing large generative VLMs. Through controlled experiments isolating the effects of attention masking, multimodal data regimes, and contrastive objectives, the authors show that bidirectional encoders outperform causal decoders for multi-vector document retrieval. Based on these insights, they train ModernVBERT, a compact 250M-parameter vision-language encoder that outperforms models up to 10x its size on ViDoRe benchmarks, with the added benefit of efficient CPU inference.

**Compliance With Llm Reviewing Policy:**

Affirmed.

**Key Questions For Authors:**

1. Have you considered or experimented with NeoBERT as an alternative text backbone? How would you expect it to compare with ModernBERT?
2. How does ColModernVBERT compare to Light-ColPali on the ViDoRe benchmarks?
3. What is the performance breakdown on ViDoRe by document type? Are there document types where the 250M model struggles relative to larger models?
4. The pixel shuffling compression ratio is adopted from prior work — did you ablate this parameter for document images specifically?

**Limitations:**

Yes

**Strengths And Weaknesses:**

## Strengths
- The iso-data ablation setup — identical architecture, data, optimization, hardware, differing only in attention masking — is exemplary. This is the right way to isolate design decisions and represents a methodological contribution beyond the specific model.
- A 250M model achieving competitive performance on document retrieval while enabling CPU inference is highly relevant for industrial deployment.
- The paper systematically examines modality alignment, contrastive training design, attention masking (enc vs. dec vs. dec-to-enc), and multi-vector vs. single-vector representations across multiple evaluation dimensions.
- Good use of the Gisserot-Boukhlef et al. (2025) controlled checkpoints to isolate the causal vs. bidirectional question, extending text-only findings to the visual domain.
- The paper is well-structured, with the ablation-then-build narrative making the design choices for ModernVBERT feel well-motivated rather than arbitrary.

## Weaknesses
- Missing related work: Light-ColPali: A concurrent approach to efficient visual document retrieval that should be discussed in related work, especially given the shared goal of compact VDR models. NeoBERT: A recent modernized BERT variant that could serve as an alternative text backbone; its omission is notable given the paper's premise of building from encoder-style models.
- The controlled enc/dec ablations use a Llama-based 210M model, but the final ModernVBERT uses a different 250M ModernBERT architecture. This architecture mismatch means the ablation conclusions may not transfer perfectly to the final model.
- Key comparisons involve small margins (e.g., ModernVBERT 68.6 vs ColPali 69.2 nDCG@5). No confidence intervals or significance tests are reported.
- While the paper evaluates on natural image retrieval and classification, the model significantly underperforms dual encoders on these tasks. The paper could better frame when ModernVBERT is and isn't the right choice.
- It is unclear whether the bidirectional advantage over causal decoders holds at larger scales (e.g., 1B+). The findings are conditioned on the ~200-250M parameter range.
- The paper uses SigLIP2-base-16b-512 exclusively. It would strengthen the findings to verify that the bidirectional advantage holds with a different vision tower.
- Where does the 250M model specifically fail compared to larger models? Understanding the performance boundary would help practitioners decide when to use ModernVBERT vs. a larger model.

---

> ### Author Rebuttal · Authors · 2026-03-31
>
> We sincerely thank the reviewer for the detailed and constructive feedback. We appreciate the recognition of both the methodological rigor of our controlled ablation setup and the practical relevance of our work for efficient deployment.
>
> Below, we address key questions and weaknesses raised in the review to respect the character limit. We are happy to address them all during the discussion phase.
>
> > Have you considered or experimented with NeoBERT as an alternative text backbone? How would you expect it to compare with ModernBERT?
>
> We have not tested NeoBERT, but we did compare with EuroBERT [1], which shares the same architecture (see [Rebuttal to Reviewer v79g](https://openreview.net/forum?id=TyVJlSHke2&noteId=7AlC7JFFXP)). Given that NeoBERT and ModernBERT perform similarly on retrieval tasks [2], we would expect comparable results for a hypothetical NeoVBERT. One practical difference worth noting: NeoBERT is only available at 250M parameters, whereas ModernBERT also offers a 150M variant, giving more flexibility along the efficiency axis.
>
> > How does ColModernVBERT compare to Light-ColPali on the ViDoRe benchmarks?
>
> Light-ColPali addresses a complementary problem: reducing the storage cost of multi-vector representations [3]. Our work focuses on reducing model size and inference cost. The two approaches are orthogonal and could be combined: applying Light-ColPali's compression on top of ColModernVBERT embeddings to reduce storage at deployment time.
>
> > What is the performance breakdown on ViDoRe by document type? Are there document types where the 250M model struggles relative to larger models?
>
> **ViDoRe v1 breakdown by split:**
>
> | Model | Late Int. | Size (B) | DocVQA | InfoVQA | TAT-DQA | arXivQA | TabFQuAD | Energy | Government | Healthcare | AI | Shift Project | Avg |
> |---|---|---|---|---|---|---|---|---|---|---|---|---|---|
> | *ColModernVBERT* (ours) | ✓ | 0.25 | 55.7 | 86.4 | 73.2 | 75.0 | 76.1 | 93.7 | 93.3 | 96.4 | 97.5 | 64.9 | 81.2 |
> | ColPali | ✓ | 2.92 | 59.1 | 82.2 | 66.2 | 80.0 | 81.9 | 92.1 | 92.8 | 94.8 | 96.9 | 70.2 | 81.6 |
>
> ColModernVBERT matches ColPali on average (81.2 vs 81.6) and outperforms it on 6 out of 8 English sub-tasks, most notably TAT-DQA (+7.0) and InfoVQA (+4.2). The only clear gaps are on the two **French** sub-tasks: TabFQuAD (−5.8) and Shift Project (−5.3), which is expected given English-only training.
>
> **ViDoRe v2 breakdown by split:**
>
> | Model | Late Int. | Size (B) | Biomedical Lectures | ESG Reports | Economics Reports | Avg |
> |---|---|---|---|---|---|---|
> | *ColModernVBERT* (ours) | ✓ | 0.25 | 55.3 | 53.3 | 60.4 | 56.0 |
> | ColPali | ✓ | 2.92 | 57.1 | 55.4 | 57.7 | 56.8 |
>
> On ViDoRe v2 ColModernVBERT is within ~2 points of ColPali on all sub-tasks.
>
> Beyond multilingual capabilities, no consistent domain-specific weakness emerges across document types. We show in our [Rebuttal to Reviewer v79g](https://openreview.net/forum?id=TyVJlSHke2&noteId=7AlC7JFFXP) that using a language backbone with stronger multilingual capacity addresses this limitation.
>
> We thank the reviewer for raising this point, and will add the breakdown in the appendix of the final version.
>
> > The pixel shuffling compression ratio is adopted from prior work — did you ablate this parameter for document images specifically?
>
> We did not ablate this parameter in terms of downstream performance, as extensive prior work already covers this design choice [4, 5]. However, we estimated the expected number of visual tokens for each shuffling ratio in Figure 12, identifying a ratio of 4 as the sweet spot, consistent with findings in the literature [4, 5].
>
> > While the paper evaluates on natural image retrieval and classification, the model significantly underperforms dual encoders on these tasks. The paper could better frame when ModernVBERT is and isn't the right choice.
>
> We do observe that a modest token-level alignment training yields strong gains for document retrieval (+6.1 nDCG@5 over the base vision encoder on ViDoRe) but does not transfer to natural image tasks, where large contrastive pretraining of dual encoders remain superior. We report and discuss this in Section 3.1, notably in lines 241–246.
>
> > Key comparisons involve small margins (e.g., ModernVBERT 68.6 vs ColPali 69.2 nDCG@5). No confidence intervals or significance tests are reported.
>
> Our main contribution is the parameter-performance trade-off (see Figure 1): we match ColPali's performance with 10× fewer parameters. Additionally, we evaluate the models on widely adopted datasets of significant size, so differences in score are meaningful. Finally, embedding-based retrieval evaluation is deterministic (up to minor hardware-level floating-point variation), so confidence intervals over repeated runs would not be informative.
>
> ---
>
> #### References
>
> - [1] https://arxiv.org/abs/2503.05500
> - [2] https://arxiv.org/abs/2502.19587
> - [3] https://arxiv.org/abs/2506.04997
> - [4] https://arxiv.org/abs/2504.05299
> - [5] https://arxiv.org/abs/2407.07726

---

> > ### Author Rebuttal · Reviewer_LN7s · 2026-04-04
> >
> > We appreciate the author's response. The main issues we raised have been sufficiently addressed.

---

### Official Review · Reviewer_jxXm · 2026-03-10

**Soundness:** 3
**Presentation:** 3
**Significance:** 3
**Originality:** 3
**Overall Recommendation:** 4
**Confidence:** 3

**Summary:**

This paper systematically investigates the key factors in designing efficient, small-scale retrieval models from scratch in the field of Visual Document Retrieval (VDR). Through a series of controlled experiments, the authors delve into the impact of core design choices—such as attention mechanisms (causal vs. bidirectional), modality alignment objectives, contrastive training data combinations, and image resolution—on downstream retrieval performance. Based on these findings, the paper proposes ModernVBERT, a 250M parameter multimodal encoder. Experiments demonstrate that its multi-vector variant, ColModernVBERT, achieves performance comparable to or approaching that of existing models with ten times the parameter size on the standard VDR benchmark (ViDoRe), while also delivering significant inference speedups on CPU. The paper presents solid work with clear contributions, offering a meaningful contribution to advancing the development of efficient multimodal retrieval models.

**Compliance With Llm Reviewing Policy:**

Affirmed.

**Final Justification:**

Thanks to the author's detailed response. Considering the overall quality of the paper, the novelty of the method, and the downstream experimental results, I have decided to maintain the current positive score.

**Key Questions For Authors:**

1. Regarding Model Scalability and Inductive Bias: Your experiments strongly demonstrate the critical importance of bidirectional attention for late interaction retrieval at the 250M parameter scale. Do you believe this inductive bias will remain crucial as models are scaled up to larger sizes (e.g., 1B+ parameters)? Or, as model capacity and data volume increase, can causal attention models potentially close this gap through scaling laws?
2. On the Mechanism of Text-Only Data Augmentation: The paper observes that incorporating text-only pairs improves visual document retrieval performance. Could you elaborate on the hypothesized mechanism behind this? Does this improvement primarily stem from enhanced query representations, or does it mainly optimize the alignment of the multimodal shared embedding space?
3. Resolution Trade-offs and Adaptive Strategies: While high-resolution inputs significantly boost document retrieval performance, they seem to cause a performance drop on natural image tasks. For practical deployment scenarios where a single model needs to handle both types of tasks, what strategy would you recommend for choosing the input resolution? In this context, do you believe VLM architectures with adaptive resolution capabilities would hold an advantage over fixed-resolution models?
4. Comparison with Model Compression Techniques: The paper compares with mainstream models up to mid-2025. To more thoroughly demonstrate the superiority of the "training from scratch" approach, I am curious how ColModernVBERT compares against state-of-the-art large models that have been distilled or quantized down to a comparable parameter size?
5. Cost-Benefit Analysis for Industrial Deployment: The paper mentions that the entire research process consumed a significant amount of compute (approximately 18k H100 GPU hours). However, according to Table 4, ColModernVBERT's score on ViDoRe v2 (eng) (56.0) remains noticeably lower than the best large models (e.g., NV-Retriever-v1's 66.3). How do the authors view this performance gap? In real-world industrial applications, is a solution that trades such a significant accuracy drop (~10 points) for improved inference efficiency considered acceptable?

**Limitations:**

yes

**Strengths And Weaknesses:**

## Strengths

1. A Valuable Research Perspective: The paper critically reflects on the current dominant paradigm of using generative VLMs for Visual Document Retrieval (VDR). By advocating for designing dedicated retrieval encoders from scratch, it successfully reduces deployment costs and improves inference efficiency. This research direction holds significant practical value for large-scale industrial applications.
2. Solid Methodological Validation: The authors employ a two-stage training strategy (modality alignment + contrastive learning) and conduct exhaustive ablation studies on key hyperparameters (such as attention mechanism, image resolution, data composition, and training steps). The experimental design is rigorous and systematic.
3. Excellent Performance-Efficiency Trade-off: ModernVBERT and its multi-vector variant achieve an outstanding balance between parameter count, performance, and inference speed. On the ViDoRe benchmark, these models, with a modest size (~250M parameters), achieve performance comparable to or even surpassing existing models with ten times the parameters, while demonstrating significant CPU inference acceleration. This strongly validates the effectiveness of the proposed approach.
4. High Reproducibility: The authors commit to open-sourcing the models, code, and data, and provide detailed experimental procedures and hyperparameter settings in the appendix. This is a positive contribution to the community, facilitating replication and future research.

## Weaknesses:
1. Limited Architectural Novelty: Although the paper advocates building VDR models "from scratch" to challenge existing paradigms, the proposed ModernVBERT architecture (an early fusion of a pre-trained text encoder and a vision tower) is not a novel design in the vision-language model (VLM) field.
2. Limited Insightfulness: Some of the paper's core conclusions (e.g., "bidirectional attention outperforms causal attention," "higher resolution improves document retrieval performance") have already been widely validated in existing research on generative VLMs. This diminishes the insightfulness of the empirical findings presented in this paper.
3. Limited Scope of Conclusions: The experiments are confined to small-parameter models (~250M), and thus cannot demonstrate whether the proposed methodology remains effective under scaling laws, which weakens its guiding significance for larger models.
4. Insufficiently Comprehensive Baselines: To more convincingly demonstrate the superiority of the "dedicated training" route over using off-the-shelf general-purpose vision encoders, the paper lacks a fair comparison with a broader range of strong, non-VDR-specific vision-text dual encoders trained on comparable data. This prevents a definitive attribution of the performance gains to the "dedicated training" approach.
5. Limited Diversity in Evaluation Tasks: The evaluation primarily focuses on document retrieval (ViDoRe) and a limited set of natural image tasks. To assess the model's claimed "general-purpose" potential, there is a lack of zero-shot or fine-tuning performance evaluations on tasks requiring fine-grained reasoning, such as Visual Question Answering (VQA), chart understanding, or document structure analysis. This makes it difficult to fully gauge the quality of its multimodal representations.
6. Incomplete Coverage of Document Scenarios: The ViDoRe benchmark may not fully represent the complexity of real-world document environments. The paper does not demonstrate the model's performance on long-tail scenarios such as low-quality scans, handwritten documents, complex multi-column layouts, or scientific literature rich with formulas, leaving its robustness in such cases unverified.

---

> ### Author Rebuttal · Authors · 2026-03-31
>
> We sincerely thank the reviewer for their thoughtful review and for recognizing the practical value, solid methodology, and strong performance-efficiency trade-off of our work. The feedback highlights several important areas, and we appreciate the opportunity to clarify our contributions and address their concerns.
> Because of the character limit, we focus on the key questions and most critical comments. We encourage the reviewer to read other reviews that may tackle other of their concerns, and will be happy to address the other points during the discussion phase
>
> > Regarding Model Scalability and Inductive Bias
>
> We agree with the reviewer that our experiments do not confirm whether the bidirectional advantage holds empirically at larger scales. Our computational budget constrained ablations to the 200–250M range, which we deliberately targeted as the most impactful regime for industrial deployment — where small models matching the performance of 10× larger ones yield the largest practical gains. However, concurrent work [1] has shown that the benefits of the bidirectional attention hold at different scales on text-only retrieval tasks, suggesting that the advantage of coupling bidirectional attention to late interaction likely persists for VDR at larger scales
>
> >Limited Diversity in Evaluation Tasks
>
> ViDoRe benchmark evaluates complex, real-world data like dense scientific formulas in ArxivQA, multi-column financial layouts, and unstructured ESG slides. While primarily focused on visual document retrieval, we also tested our model on image captioning, classification, and text retrieval to demonstrate the broader strengths and limitations of its multimodal representations.
>
> > On the Mechanism of Text-Only Data Augmentation:
>
> The inclusion of text-only pairs primarily optimizes the semantic richness and alignment of the multimodal shared embedding space. Our findings indicate that incorporating document-oriented text-text pairs matters more for downstream VDR performance than general-purpose text-image pairs. This suggests cross-modal transfer is highly effective: establishing a strong semantic representation space (even without visual inputs) is more critical for complex retrieval than general visual understanding alone.
>
>
> > Resolution Trade-offs and Adaptive Strategies:
>
> We agree with the observation that high-resolution inputs are essential for documents but can degrade natural image performance. However, in real-world industrial deployments, we generally advise against using a single unified model for both use cases.
> As demonstrated in our work, for natural images, a standard low-resolution dual encoder (like SigLIP) actually outperforms early-fusion VLMs. Deploying a heavier, slower VLM for simple natural image retrieval is computationally inefficient when a lightweight dual encoder does the job better and faster.
> If a pipeline strictly requires a single model, we recommend a smart pre-processing strategy rather than a fundamentally different architecture. A conditional processor can detect the input type (e.g., a PDF document vs. a natural photograph). If it detects a document, it upscales/resizes the image to a high resolution to preserve text; if it detects a natural image, it processes it at its native, lower resolution.
>
>
> > Comparison with Model Compression Techniques:
>
> While some recent approaches have tackled distillation on the query encoder side (e.g., NanoVDR [2]), achieving a highly performant, distilled multi-vector model for document ingestion remains an open challenge.
> Works in the text domain (such as ColBERTZero [3]) indicate that this is a promising direction, but applying these techniques to multimodal VDR models is largely left for future work. We believe our "training from scratch" approach provides a robust foundation for compact models prior to any distillation efforts.
>
> > Cost-Benefit Analysis for Industrial Deployment:
>
> ColModernVBERT is designed to be Pareto optimal for specific deployment constraints.
> While there is a performance gap with state-of-the-art multi-billion parameter models, ColModernVBERT uniquely enables highly efficient inference on consumer CPUs, achieving more than a 7x speedup over comparable models.
> Furthermore, recent larger multi-vector models (such as  NV-Retriever-v1 [4]) often output very high embedding dimensions per token, which drastically inflates vector database storage costs. For industrial pipelines bottlenecked by indexing latency, CPU availability, and storage space, accepting an accuracy trade-off for massive operational efficiency is a highly desirable and practical solution.
>
>
> ---
>
> **References**
> - [1] https://arxiv.org/abs/2506.23115
> - [2] https://arxiv.org/abs/2603.12824
> - [3] https://arxiv.org/abs/2602.16609
> - [4] https://arxiv.org/abs/2507.05513

---

> > ### Author Rebuttal · Reviewer_jxXm · 2026-04-03
> >
> > The authors have addressed the majority of my concerns constructively.
> >
> > Although Weak#1(Model Scalability) does not include additional quantitative results to address the issue directly, given the time and computational constraints, I think this is acceptable.

---

### Official Review · Reviewer_v79g · 2026-03-11

**Soundness:** 3
**Presentation:** 3
**Significance:** 3
**Originality:** 3
**Overall Recommendation:** 4
**Confidence:** 2

**Summary:**

This paper systematically revisits design choices in Visual Document Retrieval (VDR) — a paradigm in which document pages are embedded as images and matched against user queries, serving as the first retrieval stage in Retrieval-Augmented Generation (RAG) pipelines. The authors outline a general aspect of the VDR training pipeline by isolating the impact of key factors: attention masking schemes (causal vs. bidirectional), modality alignment objectives (CLM vs. MLM), image resolution, and contrastive training data regimes. The article presents an important concept: that causal attention, inherited from generative VLMs, is fundamentally suboptimal for multi-vector (late interaction) retrieval, while bidirectional encoders unlock substantial gains in this setting. Guided by these findings, the authors train ColModernVBERT, a compact 250M-parameter vision-language encoder that matches or outperforms models up to 10× its size on the ViDoRe benchmark, while enabling efficient CPU-based inference.

The main contributions of this paper are:

1. A systematic ablation study isolating the effect of attention masking, modality alignment objectives, image resolution, and contrastive data composition on VDR performance.
2. The finding that bidirectional attention provides a large benefit specifically in multi-vector/late-interaction settings (+10.6 nDCG@5), while having minimal impact on single-vector retrieval.
3. ColModernVBERT, a 250M-parameter model that achieves competitive performance with state-of-the-art models that are significantly larger, with substantially lower inference latency (including on CPU hardware).

**Compliance With Llm Reviewing Policy:**

Affirmed.

**Final Justification:**

Thanks for the response. The authors' rebuttal effectively addressed the raised issues. As the core strengths of the paper remain well-supported, I am pleased to maintain my positive assessment.

**Key Questions For Authors:**

1. Can you provide more insight into why bidirectional attention helps late interaction so much more than single-vector retrieval? Is there an analysis of how MaxSim scores distribute differently between encoder and decoder models?

2. The dec-to-enc model (converting a causal decoder to bidirectional attention mid-training) does not recover the performance of a natively trained encoder. Have you investigated at what point in training this transition becomes too costly to recover from?

3. Have you evaluated ColModernVBERT on any multilingual VDR benchmarks, even informally? Given that the base encoder is English-only, how sensitive do you expect the architecture to be to extending to other languages?

**Limitations:**

yes

**Strengths And Weaknesses:**

Soundness: The flow from ablation experiments to the final model design is logical and convincing. Each experimental result directly informs a design decision in ModernVBERT. However, while the experimental finding that bidirectional attention substantially boosts late-interaction performance is well-supported empirically, the paper offers limited intuition for why this is the case beyond a brief mention that causal decoders cannot correctly contextualize earlier tokens. A more thorough analysis (e.g., visualizing attention patterns, or discussing how MaxSim interacts with unidirectional context) would strengthen the paper considerably.


Presentation: The paper assumes familiarity with several concepts that are not introduced: causal vs. bidirectional attention masks, vision towers, nDCG as an evaluation metric, and the general role of VLMs in RAG pipelines. A high-level diagram illustrating the full VDR pipeline (including where each design choice fits) would substantially improve readability. Figure 2 partially addresses this but does not explain the architecture within the broader RAG context. Additionally, nDCG@5 is used throughout as the primary metric but is never defined or contextualized, making it difficult to interpret the magnitude of reported gains.


Significance: Demonstrating that strong VDR performance is attainable at 250M parameters with CPU-compatible inference is valuable for real-world RAG deployments with constrained hardware. The result that text-only retrieval pairs can improve visual document retrieval performance (Section 3.2) is also a useful and transferable insight for the field.


Originality: The paper correctly identifies that no prior work has systematically studied training natively bidirectional encoders for VDR, and fills this gap with controlled experiments.

---

> ### Author Rebuttal · Authors · 2026-03-31
>
> We sincerely thank the reviewer for their constructive feedback and for recognizing the value of our contributions including the systematic demonstration of the advantages of bidirectional attention for multi-vector retrieval, the significance of the performance of our 250M parameter model, and the transferability of our insights. We address each comment and question below.
>
> > Soundness
>
> We appreciate the feedback. Conceptually, causal attention masks strictly limit the flow of information between token representations (e.g., the first token can only attend to itself). While this unidirectional context is a necessity for generative tasks, it severely restricts the representational power of individual tokens for embedding tasks. In a Late Interaction (multi-vector) setting—which relies entirely on fine-grained, token-level matching—this restriction bottlenecks representation richness without offering any positive trade-off. Our  results confirm similar trends observed in related work on text representation tasks [1]. We agree this is an important discussion and will use the extra page limit in the final version to expand on this conceptual explanation.
>
> > Presentation
>
> We appreciate the reviewer highlighting where additional context could benefit readers. We note that several of these concepts are discussed in the paper: the role of VLMs in RAG pipelines is introduced in lines 39–48, and causal and bidirectional attention are formally defined through their respective loss objectives in Equations 1 and 2, with an extensive discussion available in the related work we build upon [1]. That said, we agree redefining some of these could be made more accessible. In the final version, we will add a self-contained appendix defining the attention schemes and nDCG for readers less familiar with these areas. We will also replace "vision tower" with the more explicit "vision encoder" throughout the paper. Finally, we will add references to RAG architectures employing VDR (such as [2], [3]) to better situate our work in the broader pipeline context.
>
> > Question 1
>
> Our primary focus was on establishing the empirical evidence necessary to build a strong, compact VDR model, and therefore, we have not conducted a full MaxSim score distribution analysis in this work.
> Conceptually, decoder models are forced to compress the entirety of a sequence's information into the final tokens, as only those end tokens have access to the full context. It would remain an interesting future analysis to better analyze the fine-grained mechanisms behind the MaxSim scoring function. Recent work provides insights into these fine-grained behaviors by analyzing MaxSim distributions, specifically within the text domain [5].
>
> > Question 2
>
> For the annealing experiments, we utilized a dec-to-enc checkpoint provided by Gisserot-Boukhlef et al. [1] to compare against the other variants. In their own analysis (Section 4 of their paper), they demonstrate that the ideal ratio of CLM to MLM training phases for textual retrieval falls between 25% and 75%. Because the specific dec-to-enc checkpoint we used did not yield competitive performance after the modality alignment phase, conducting a full ablation to find the exact optimal transition point fell outside the primary scope and compute budget of this specific study.
>
> > Question 3
>
> We have indeed evaluated our model on the multilingual splits of the ViDoRe v2 benchmark. As anticipated, because ColModernVBERT relies on an English-only base text encoder, its performance naturally degrades on non-English tasks. To further analyze multilingual approaches, we trained a parallel variant, ColEuroVBERT, by simply swapping the English-only text backbone for a natively multilingual one (EuroBERT [4]).
> As shown in the table below, this architectural swap significantly improves performance across non-English languages (French, Spanish, German) and drastically reduces cross-lingual variance, with the expected trade-off in peak English performance:
> | Model | English | French | Spanish | German |
> | :--- | :---: | :---: | :---: | :---: |
> | ColModernVBERT | **0.560** | 0.376 | 0.277 | 0.225 |
> | ColEuroVBERT | 0.483 | **0.389** | **0.425** | **0.342** |
>
> This demonstrates that the architectural benefits we identified (such as the synergy between bidirectional attention and multi-vector matching) are not language-bound. These results hint that extending our approach to new languages could be achieved using a multilingual text backbone, or other techniques such as cross-lingual distillation. We will include these multilingual findings in the appendix of the final version.
>
> We thank the reviewer once again for their valuable feedback.
>
> ---
>
> **Citations** (We put ArXiv links instead of full citations to respect the character limit.)
>
> [1] https://arxiv.org/abs/2507.00994
>
> [2] https://arxiv.org/abs/2502.16636
>
> [3] https://arxiv.org/abs/2410.10594
>
> [4] https://arxiv.org/abs/2503.05500
>
> [5] https://arxiv.org/abs/2603.26259

---

> > ### Author Rebuttal · Reviewer_v79g · 2026-04-04
> >
> > I thank the authors for their response, which has addressed my concerns sufficiently.

---

### Official Review · Reviewer_BA3V · 2026-03-17

**Soundness:** 3
**Presentation:** 3
**Significance:** 3
**Originality:** 2
**Overall Recommendation:** 4
**Confidence:** 4

**Summary:**

This paper revisits the design of Visual Document Retrieval (VDR) models and argues that current approaches which largely adapted from generative vision-language models are suboptimal, especially for multi-vector retrieval. Through systematic analysis of training factors such as attention masking, multimodal data, and contrastive objectives, the authors identify key limitations of generative paradigms. Based on these insights, they propose ModernVBERT, a compact 250M-parameter vision-language encoder that achieves strong retrieval performance, outperforming significantly larger models while enabling efficient CPU-based inference.

**Compliance With Llm Reviewing Policy:**

Affirmed.

**Final Justification:**

Thanks for the rebuttal. The authors have addressed the majority of my concerns constructively. I have improved my final rating to WA.

**Key Questions For Authors:**

Please refer to the weaknesses.

**Limitations:**

yes

**Strengths And Weaknesses:**

Strengths:

1. The paper critically revisits the common practice of adapting generative VLMs for retrieval, providing a clear and timely analysis of their limitations.
2. The authors conduct controlled experiments to isolate the effects of key factors (e.g., attention masking, data regimes, contrastive objectives), offering valuable insights for the community.
3. The proposed ModernVBERT demonstrates that a relatively small 250M encoder can outperform much larger models, highlighting the effectiveness of retrieval-oriented design.
4. The model enables CPU-based inference with competitive performance, which is highly relevant for real-world deployment scenarios.

Weaknesses:
1. While the empirical study is thorough, the core model (encoder-based architecture) is conceptually straightforward, and the novelty mainly lies in careful design and analysis.
2. The conclusions about generative models being suboptimal may depend on specific training setups, and broader validation across more diverse configurations would strengthen the claim.
3. The evaluation focuses on document retrieval benchmarks; it remains unclear how well the findings generalize to other multimodal retrieval or reasoning tasks.
4. It is not fully clear whether all compared generative models are equally optimized or tuned for retrieval, which could affect the strength of the conclusions.
5. Although the paper highlights issues in multi-vector settings, deeper analysis of why generative models fail (e.g., representation structure) could further strengthen the work.
6. Missing related work of NL-DIR (Towards natural language-based document image retrieval: new dataset and benchmark, CVPR 2025)

---

> ### Author Rebuttal · Authors · 2026-03-31
>
> We thank the reviewer for the thoughtful review and constructive feedback which recognizes the timeliness, clarity and practical relevance of our work, as well as the value of the uncovered insights and the effectiveness of the openly released artefacts.
>
> > While the empirical study is thorough, the core model is conceptually straightforward, and the novelty mainly lies in careful design and analysis.
>
> While the architecture in itself is not particularly novel, this work first shows MLM training can be used for VLM training and examines the impact of bidirectional attention in this sort of early fusion model. Before this work, it was non-obvious that such performance could emerge on models of small size (<1B), which opens the way to new applications. As stated by the reviewer in the strengths, these findings go along many non-trivial and carefully ablated training and data design choices.
>
> > The conclusions about generative models being suboptimal may depend on specific training setups, and broader validation across more diverse configurations would strengthen the claim.
>
> The claim made is that current SOTA visual document embedders are repurposed generative models, but that some design principles can be optimized when designing an embedding model. Typically, we show that bidirectional attention unlocks the full potential of late interaction, we explore the data regimes necessary to improve modality alignment showing that documents require longer modality alignment than for natural images, ablate contrastive training length, ablate data mixes, image resolutions, matching functions, and evaluate performance on various types of data and tasks  (document retrieval, natural image retrieval, zero-shot  and light training classification tasks, text retrieval on nanoBEIR). All in all, several dozens of varied training configurations and multiple evaluation benchmarks are used to draw robust conclusions.
>
> To further demonstrate the general nature of our conclusions, we have aligned a different LLM backbone (EuroBert), pretrained on another language distribution and architecture. Our recipe directly applies, and further shows multilingual capabilities from the backbone transfer to multilingual retrieval. These results will be added to the appendix and are further detailed in [rebuttal v79g, Q3](https://openreview.net/forum?id=TyVJlSHke2&noteId=PGtkco3YBY).
>
> > The evaluation focuses on document retrieval benchmarks; it remains unclear how well the findings generalize to other multimodal retrieval or reasoning tasks.
>
> While the primary focus is Visual Document Retrieval, the evaluation is not restricted to document retrieval alone. Section 2.3 also details the evaluation that includes natural-image retrieval as well as zero-shot and fine-tuned classification tasks across multiple domains (results in Figure 3, Table 2), with further details in Section C.5. of the appendix. We also include text-to-text retrieval on the NanoBeir dataset in the appendix (Table 11), showcasing ModernVBert outperforms strong text-only baselines such as Qwen3-Embedding-0.6B.
>
> > It is not fully clear whether all compared generative models are equally optimized or tuned for retrieval, which could affect the strength of the conclusions.
>
> The main ablation study was explicitly designed to minimize tuning-related confounders. The ablation models are matched in architecture, data, and pretraining setup, with the same batch sizes, optimizers, schedulers, and hardware. Learning rates of all model variants and baselines are further heavily tuned, which represented a significant fraction of our compute budget. We will add a section in the appendix to detail our tuning process and list final values chosen.
>
> > Although the paper highlights issues in multi-vector settings, deeper analysis of why generative models fail (e.g., representation structure) could further strengthen the work.
>
> Rather than claiming causal models “fail,” we argue they are less suited to representation learning, especially in late-interaction settings. The paper shows that while single-vector results are close between encoder and decoder variants, bidirectional attention yields a large boost for late interaction matching (+10.6 nDCG@5). The manuscript also gives the core intuition for this gap: in causal models, tokens appearing early in the sequence cannot be fully contextualized, which is especially detrimental for fine-grained token-level matching (lines 260–274, right column). We will expand this discussion in the final version.
>
> > Missing related work of NL-DIR (Towards natural language-based document image retrieval: new dataset and benchmark, CVPR 2025)
>
> We will add NL-DIR to the related work and clarify its relationship to our evaluation suite. In practice, NL-DIR has a large document overlap with the DocVQA split of the ViDoRe dataset, which means it is not orthogonal from our current evaluation setup.
>
>
>
> We thank the reviewer once more for their feedback.

---

> > ### Author Rebuttal · Reviewer_BA3V · 2026-04-03
> >
> > Thanks for the rebuttal. The authors have addressed the majority of my concerns constructively.

---

### Official Review · Reviewer_F5ia · 2026-03-22

**Soundness:** 3
**Presentation:** 4
**Significance:** 4
**Originality:** 3
**Overall Recommendation:** 5
**Confidence:** 5

**Summary:**

The paper investigates the design space of Visual Document Retrieval (VDR) models, specifically challenging the current trend of repurposing large, generative vision-language decoders for embedding tasks. The authors systematically analyze key training factors, including attention masking (causal vs. bidirectional), modality alignment scaling, and contrastive data regimes.
Their primary finding is that bidirectional attention is more effective than causal attention for late-interaction (multi-vector) retrieval, which is the current state-of-the-art approach for document search. Based on these insights, the authors develop and release ModernVBERT, a 250M-parameter encoder-based multimodal model. Despite its small size, the model outperforms recent retrievers up to 10 times its size and enables efficient inference on CPU hardware, making it suitable for industrial RAG applications.

**Compliance With Llm Reviewing Policy:**

Affirmed.

**Final Justification:**

Thanks for the response. The authors' rebuttal effectively addressed the raised issues. As the core strengths of the paper remain well-supported, I am pleased to maintain my positive assessment.

**Key Questions For Authors:**

1. Does the observed bidirectional advantage for Late Interaction persist at the multi-billion parameter scale, or is it a phenomenon specific to smaller encoders?
2. What drives the classification degradation at 2048px resolution? Is this a fundamental bottleneck in the attention mechanism's ability to process global vs. local features?
3. Why are document-centric representations significantly more sensitive to parameter interpolation (e.g., SLERP) than generalist visual embeddings?

**Limitations:**

Yes, the authors have adequately discussed their limitations, including the focus on relatively small models , the current restriction to the English language , and the environmental costs associated with training.

**Strengths And Weaknesses:**

Soundness
1. Strength: The experimental design is rigorous, leveraging iso-data controlled settings to isolate the effects of key variables (e.g., attention masks, LM objectives). Evaluation spans diverse benchmarks, offering a comprehensive view of performance trade-offs.
2. Weakness: Core comparisons are limited to the 210M–250M scale. It remains unclear whether the observed bidirectional advantages generalize to multi-billion-parameter regimes where current SOTA models operate.

Presentation
1. Strength: The paper is well-structured and clearly written. Visualizations (e.g., Fig. 2 for early fusion, Fig. 4 for mask comparison) effectively illustrate key insights. The methodology section provides sufficient detail (hyperparameters, data mixtures, training stages) to support reproducibility.

Significance
1. Strength: The work targets a critical bottleneck in industrial RAG systems, namely the latency and hardware cost of VDR models. The finding that retrieval performance benefits more from LM alignment than high-level visual classification offers actionable guidance for retriever design.

Originality
1. Strength: Proposes a native bidirectional visual retriever via a systematic “ground-up” design, departing from the dominant decoder-only VLM paradigm.
2. Strength: Introduces the NatCap dataset, which leverages synthetic captions to mitigate data scarcity in contrastive learning and improve cross-modal transfer.
3. Weakness: The architecture primarily integrates existing components (e.g., ModernBERT backbone, SigLIP2 vision encoder), limiting architectural novelty.

---

> ### Author Rebuttal · Authors · 2026-03-31
>
> We thank the reviewer for the detailed review and the insightful questions raised. We appreciate the recognition of our experimental rigor, practical significance for industrial RAG, and the originality of departing from the decoder-only paradigm.
>
> Below, we address each question raised in the review:
>
> > Does the observed bidirectional advantage for Late Interaction persist at the multi-billion parameter scale, or is it a phenomenon specific to smaller encoders?
>
> We agree with the reviewer that our experiments cannot confirm whether our findings holds empirically at larger scales. Our budget constrained our experiments to the 200–250M range, which we targeted as more impactful regime for industrial deployment than >1B. However, concurrent work [1] has shown that the benefits of the bidirectional attention hold at different scales on text-only retrieval tasks, suggesting that the advantage of coupling bidirectional attention to late interaction likely persists for VDR at larger scales.
>
> >What drives the classification degradation at 2048px resolution? Is this a fundamental bottleneck in the attention mechanism's ability to process global vs. local features?
>
> We believe this is driven by the representation itself rather than an attention bottleneck. Most natural image classification and captioning datasets contain images well below 512px in resolution. Upscaling them to 2048px does not add meaningful information: local patches become noisy and redundant, forcing the model to attend over a much longer sequence without additional signal. Document images exhibit the opposite behavior: they are natively high-resolution with dense local information (text, charts, fine-grained layouts), so downscaling actively destroys content the vision encoder needs. This asymmetry, visible in Table 2, is probably not a fundamental limitation of the attention mechanism, but a consequence of applying a resolution regime optimized for one modality to another. Treating each document source with the proper resolution is crucial, hybrid processors or models becomes essential.
>
> > Why are document-centric representations significantly more sensitive to parameter interpolation (e.g., SLERP) than generalist visual embeddings?
>
> As shown in Figure 11, SLERP interpolation between the base model and its contrastively fine-tuned variant improves image classification (+6.1) but degrades document retrieval (-5.2 vs. base+CL). Our interpretation is that contrastive fine-tuning on the document retrieval mixture specializes the model to this information-heavy modality, so the learned features diverge from the generalist pretraining manifold. Interpolating between these two misaligned regions lands in a noisier middle ground. By contrast, for image classification, contrastive training acts more as a refinement of the pretraining visual representations, so the interpolation path remains meaningful.
>
> ---
>
> #### **Resources**
> [1] O. Weller et al., ‘Seq vs Seq: An Open Suite of Paired Encoders and Decoders’, 2025.

---

> > ### Author Rebuttal · Reviewer_F5ia · 2026-04-04
> >
> > We acknowledge the authors' rebuttal and appreciate their response. The main issues we raised have been clearly and sufficiently addressed.

---

### Decision · Program_Chairs · 2026-04-30

**Decision:**

Accept (regular)

**Comment:**

ModernVBERT provides a technically sound and well-written case for small, bidirectional encoder-based models in Visual Document Retrieval, outperforming models ten times its size on the ViDoRe benchmark. Reviewers strongly praised the systematic analysis and practical efficiency for constrained hardware, and the authors’ outstanding rebuttal fully resolved all initial concerns. This valuable contribution clearly warrants an Accept recommendation.